# A Proximal-Sinkhorn-Newton Method for Entropic Optimal Transport

## Abstract

Entropic optimal transport (OT) enables efficient distribution alignment through the Sinkhorn method. However, it suffers from numerical instability and slow convergence under weak entropic regularization. We propose a two-stage framework that establishes an inexact-to-exact paradigm to address these challenges. The first stage employs an inexact proximal point method to decompose the entropic OT into simpler subproblems, yielding approximate solutions with superior numerical stability. The second stage employs a sparse Newton method with global convergence and a locally quadratic rate to refine the approximate solutions. Compared to previous Newton-based algorithms, it accelerates updates and prevents the objective value from plateauing during optimization. With numerical instability handled in the first stage, Sinkhorn scaling can provide an alternative to the Newton method under relatively heavy entropic regularization. The resulting Proximal-Sinkhorn-Newton method enjoys the strengths of three approaches and outperforms the baselines across various regularizations and error tolerances.

## 1 Introduction

Optimal transport (OT) seeks the best transportation plan from an ensemble of sources to targets (Linial et al., 1998; Peyré et al., 2017). It is an important task in machine learning, such as GAN (Arjovsky et al., 2017; Genevay et al., 2018), graph matching (Shen et al., 2024), domain adaptation (Nguyen et al., 2021; Turrisi et al., 2022), statistical learning (Oneto et al., 2020; Huynh et al., 2020), robust and efficient unsupervised learning (Lei et al., 2019; Onken et al., 2021), feature selection (Cao and Zhang, 2024; Cao et al., 2026), and so on. For an overview of further applications of optimal transport, readers are referred to (Peyré and Cuturi, 2019).

In this work, we focus on the entropic optimal transport problem of the form

$$P_\beta = \arg\min_{P \in \mathbb{U}} C \cdot P + \frac{1}{\beta}\mathcal{H}(P), \ \ \mathcal{H}(P) := P \cdot \mathring{\log} P \tag{1}$$

where $P \in \mathbb{R}_+^{n \times n}$ represents a transport plan, $\mathbb{U} := \left\{ P \in \mathbb{R}_+^{n \times n} : P\mathbf{1} = \boldsymbol{a}, P^\top \mathbf{1} = \boldsymbol{b} \right\}$ is the feasible region where $\boldsymbol{a}, \boldsymbol{b} \in \mathbb{R}_+^n$ are the source and target distributions respectively, $C \in \mathbb{R}^{n \times n}$ is a cost matrix, $C \cdot P = \sum C_{ij} P_{ij}$ is the transport cost, $\beta > 0$ is the regularization parameter, $\mathcal{H}(\cdot)$ is the negative entropy, and $\mathring{\log}$ denotes the element-wise logarithm operator i.e., $P \cdot \mathring{\log} P = \sum P_{ij} \log(P_{ij})$. Cuturi (2013) demonstrates that the solution to (1) admits a closed-form characterization:

$$P_\beta = \Gamma_{\mathbb{U}}^{\mathrm{kl}} \left( \mathring{\exp}(-\beta C) \right) = D_{(\boldsymbol{u})} \, \mathring{\exp}(-\beta C) \, D_{(\boldsymbol{v})}, \tag{2}$$

where $\Gamma_{\mathbb{U}}^{\mathrm{kl}}(\cdot)$ denotes the projection onto the set $\mathbb{U}$ with respect to the Kullback–Leibler (KL) divergence (Peyré et al., 2017), $\mathring{\exp}(\cdot)$ represents element-wise exponential operator, $D_{(\boldsymbol{u})}$ and $D_{(\boldsymbol{v})}$ are diagonal matrices constructed from positive scaling vectors $\boldsymbol{u}, \boldsymbol{v} \in \mathbb{R}_+^n$. The two vectors can be computed iteratively via the classical Sinkhorn algorithm (Sinkhorn, 1964; Cuturi, 2013):

$$K = \mathring{\exp}(-\beta C), \quad \boldsymbol{v}^{(l+1)} = \boldsymbol{a} \oslash (K\boldsymbol{u}^{(l)}), \quad \boldsymbol{u}^{(l+1)} = \boldsymbol{b} \oslash (K^\top \boldsymbol{v}^{(l+1)}). \tag{3}$$

where $\oslash$ is the element-wise division.

According to (2), solving the entropic OT relies on two components: the kernel matrix $K$ and the scaling vectors $\boldsymbol{u}, \boldsymbol{v}$. These introduce two intrinsic computational challenges for entropic OT with a

large $\beta$: (a) numerical instability caused by division by zero, which can occur when $K$ contains an entire row of zeros; and (b) the iterative computation of $\boldsymbol{u}$ and $\boldsymbol{v}$ with complexity tied to $\beta$. These challenges are inherent to the entropic OT and continue to motivate ongoing research.

**Numerical instability**. The stable Sinkhorn (Benamou et al., 2015) addresses the numerical instability by performing computations in log-space, but introduces considerable overhead. The proximal point method (POT) (Xie et al., 2020) generates a sequence $\{P^{(t)}\}_{t=1,2,\dots}$ to obtain the solution to OT by solving a series of entropic OT problems with small values of $\beta$. To improve efficiency, the inexact proximal point method (IPOT) (Xie et al., 2020) computes the proximal operator approximately at each iteration, yet $\{\hat{P}^{(t)}\}_{t=1,2,\dots}$ still converges to the exact solution (shown in Figure 1). However, the algorithm exhibits only linear convergence in OT and does not benefit from the efficiency of entropic OT, thus becoming potentially expensive for large-scale problems.

**Computational complexity**. The Sinkhorn algorithm enjoys an $\mathcal{O}(n^2)$ per-iteration cost, but due to its sub-linear convergence, the algorithm may require a prohibitively large number of iterations to attain a moderately high accuracy (Tang and Qiu, 2024). The Newton method has a local quadratic convergence rate, while even a single Newton step requires $\mathcal{O}(n^3)$ cost (Brauer et al., 2017). The Sinkhorn-Newton-Sparse (SNS) algorithm (Tang et al., 2024) sparsifies the Hessian matrix to reduce the computational cost to $\mathcal{O}(n^2)$. However, its warm start may require hundreds of Sinkhorn iterations (as shown in Figure 1). Tang and Qiu (2024) propose an SSNS to overcomes this problem, whereas the objective value may experience a period of stagnation during the Newton iteration.

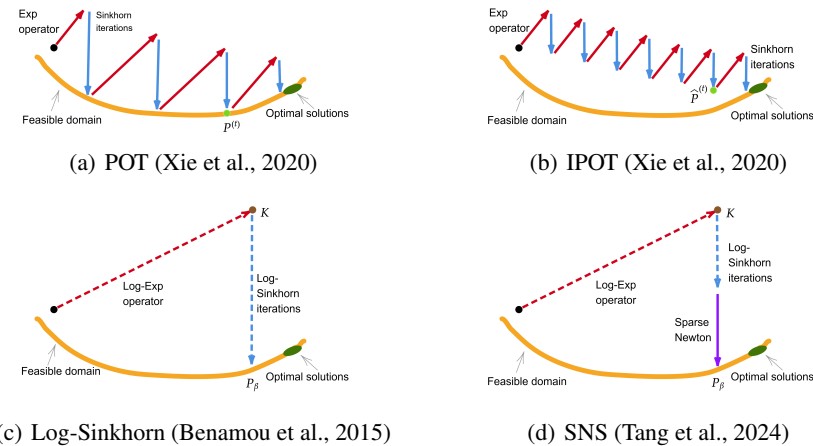

(a) POT (Xie et al., 2020)                  (b) IPOT (Xie et al., 2020)

(c) Log-Sinkhorn (Benamou et al., 2015)      (d) SNS (Tang et al., 2024)

Figure 1: Schematic of the path of methods. The distance is in the Bregman sense. POT and IPOT are designed for OT. In fact, they require far more iterations than those illustrated in the figure to converge; log-Sinkhorn and SNS are designed for entropic OT.

In light of the above challenges, a natural question arises:

> How to design an entropic optimal transport algorithm that maintains stability and efficiency under varying entropic regularization parameters and error tolerances?

A promising approach is to weave complementary algorithms into a single framework that leverages their respective strengths. Specifically, we employ the inexact proximal point method to stably construct a transport plan. If this plan does not meet the desired accuracy, it can serve as a high-quality initializer for faster secondary solvers. Such an idea is supported by the following theorem.

**Theorem 1** *Inexact-to-Exact Transition*
*If $\hat{P}^{(t)}$ denotes the transport plan updated at the $t$-th iteration in the inexact proximal point method with a fixed proximal parameter $\Delta\beta$, then*

$$P_\beta = \Gamma_{\mathbb{U}}^{\mathrm{kl}}(\hat{P}^{(t)}), \; where \; \beta = t\Delta\beta. \tag{4}$$

This theorem establishes a fundamental connection between the inexact proximal point method and the entropic OT. Specifically, it allows the inexact iterate $\hat{P}^{(t)}$ to be refined through the KL-projection into the exact solution of the entropic OT problem. Interpreting $\hat{P}^{(t)}$ as a processed kernel matrix reveals two distinct benefits over the naive counterpart $K$: improved numerical robustness and reduced distance to the solution $P_\beta$ (shown in Figure 1). These properties make it a more desirable input for subsequent refinement algorithms such as the Sinkhorn scaling or the Newton method.

The contributions of this paper are threefold:

- **Numerical instability: addressed via an inexact proximal point framework**. We use a proximal point method that safely decomposes the problem into a sequence of easier and safer entropic OT problems. By employing an inexact proximal operator, we derive an Inexact Proximal point method for Entropic OT (IP-EOT), stably and efficiently producing an approximate solution that other methods can further refine.

- **Computational complexity: reduced via a sparse Newton method**. We introduce a novel sparse Newton method with global convergence and locally quadratic rates. The positive definiteness of the sparsified Hessian matrix enables the use of efficient incomplete Cholesky conjugate gradient solvers to accelerate each Newton update. Furthermore, we integrate line search with Sinkhorn scaling to prevent the objective value from plateauing during the optimization process.

- **Adaptive hybrid solver**. Our framework leverages IP-EOT either as a low-accuracy standalone solver or as a high-accuracy warm-start for both Sinkhorn and Newton methods, thereby overcoming instability of the Sinkhorn algorithm and expensive initialization of the Newton method. A proposed adaptive criterion selects the secondary solver by exploiting the Hessian's sparsity dependence on $\beta$: the Newton method is preferred for large $\beta$ (sparser Hessian), while the Sinkhorn scaling is chosen for smaller $\beta$.

In short, we address two fundamental challenges in entropic optimal transport—numerical instability and high computational complexity—by introducing a unified inexact-to-exact framework, termed Proximal-Sinkhorn-Newton (PSN), which intelligently integrates complementary algorithms. The overall architecture of PSN is illustrated in Figure 2.

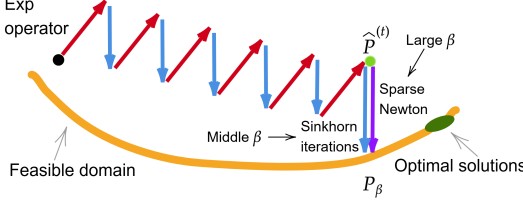

Figure 2: The trajectory of PSN consists of two stages: (a) IP-EOT for an efficient and stable approximation $\hat{P}^{(t)}$, and (b) KL-projection by either Sinkhorn scaling or the Newton method for refinement.

## 2 RELATED WORK

**Computational Optimal Transport.** Since the Sinkhorn algorithm (Cuturi, 2013) made entropic optimal transport practical, researchers have polished it from every angle. First-order alternating minimisation (Guminov et al., 2021), Greenkhorn (Altschuler et al., 2017) and their greedy variants speed up the basic matrix-scaling loops. Adaptive primal-dual strategies, such as the similar-triangle method (Dvurechensky et al., 2020) and accelerated mirror descent (Li et al., 2025), dynamically tune step-sizes to reduce the iteration count. When high accuracy is required, Newton-type solvers exploit second-order information to yield local quadratic convergence (Brauer et al., 2017; Tang et al., 2024; Tang and Qiu, 2024). Yet, an efficient method applicable to a wide range of $\beta$ is still lacking, which motivates this work.

**Newton method.** The Newton method achieves rapid convergence, whereas handling a dense Hessian incurs $\mathcal{O}(n^3)$ complexity (Powell and Toint, 1979). To mitigate this, Tang et al. (2024); Tang and

Qiu (2024); Kemertas et al. (2025) sparsify the Hessian matrix to reduce the per-iteration cost to approximately $\mathcal{O}(n^2)$. Iterative solvers such as Conjugate Gradient (CG) (Golub and Van Loan, 2013) benefit significantly from preconditioning (Caraba, 2008), yet this has been overlooked in prior work on optimal transport. We therefore apply a preconditioner to accelerate the Newton method.

**Proximal point method.** The proximal point method transforms complex optimization problems into a sequence of simpler subproblems by introducing a proximal operator (Rockafellar, 1976). Based on the Bregman distances, Chen and Teboulle (1993) formalized the Bregman proximal point algorithm and established its convergence. Recently, this method has been applied to OT, substantially reducing computational complexity (Xie et al., 2020; Chu et al., 2023; Chizat, 2024). The unclear trajectory of the regularization parameter during iteration has hindered use of the proximal point method in entropic OT, which our work seeks to address.

## 3 PROXIMAL POINT METHODS FOR ENTROPIC OPTIMAL TRANSPORT

We propose an Inexact Proximal point method for Entropic OT (IP-EOT) that yields an approximation $\hat{P}_\beta$ by explicitly incorporating $\beta$ into the proximal step. This approximation can also be derived from an approximate KL-projection $\hat{\Gamma}_{\mathbb{U}}^{\mathrm{kl}}(\overset{\circ}{\exp}(-\beta C))$, a typical instance of which is early termination of Sinkhorn scaling. Furthermore, we demonstrate that $\hat{P}_\beta$ can be refined into the exact solution $P_\beta$ through an exact KL-projection.

**Proximal point methods** are iterative algorithms for convex optimization, well-suited for nonsmooth or composite objectives. Their core principle involves solving a regularized subproblem at each step. This process optimizes the objective while ensuring the stability of the iterative sequence. To solve the OT (without the entropic term), the proximal point method reads as follows

$$P^{(t+1)} = \arg\min_{P \in \mathbb{U}} C \odot P + \beta^{(t+1)} \mathcal{B}(P, P^{(t)}), \tag{5}$$

$$\mathcal{B}(P, P^{(t)}) = P \cdot \overset{\circ}{\log}(P \oslash P^{(t)}) - (P - P^{(t)}) \cdot \mathbf{1}\mathbf{1}^\top, \tag{6}$$

where $\odot$ is the element-wise product. $\mathcal{B}(\cdot, \cdot)$ is the Bregman divergence used to measure the distance between two probability vectors/matrices (Benamou et al., 2015). This formula can be equivalently rewritten as

$$P^{(t+1)} = \arg\min_{P \in \mathbb{U}} C^{(t)} \odot P + \beta^{(t+1)} \mathcal{H}(P), \quad C^{(t)} = C - \beta^{(t+1)} \overset{\circ}{\log} P^{(t)}. \tag{7}$$

The update can be computed by applying a KL-projection to a modified kernel matrix, yielding:

$$P^{(t+1)} = \Gamma_{\mathbb{U}}^{\mathrm{kl}} \left( \overset{\circ}{\exp}(-C\beta^{(t+1)}) \odot P^{(t)} \right). \tag{8}$$

As $t \to \infty$, $P^{(t)}$ converges to the optimal transport plan. To improve efficiency, Xie et al. (2020) proposed the inexact proximal point method (IPOT), which performs only one Sinkhorn iteration to approximate the projection in (8). The resulting sequence $\{\hat{P}^{(t)}\}_{t=1,2,\ldots}$ can efficiently converge to the optimal transport plan.

**Inexact proximal point method for entropic OT**. Theorem 1 allows the inexact proximal point method to efficiently offer a $\hat{P}_\beta$. This theorem relies on explicitly tracking the regularization parameter $\beta$ across proximal iterations, as detailed below.

**Lemma 1 (Explicit form of $\boldsymbol{\beta}$)** *The iterative schemes of the exact/inexact proximal point method for entropic OT admit the following explicit representation:*

$$P^{(t)} = \Gamma_{\mathbb{U}}^{\mathrm{kl}}(\bigodot_{i=1}^{t} \overset{\circ}{\exp}(-C\beta^{(i)})), \quad \hat{P}^{(t)} = \hat{\Gamma}_{\mathbb{U}}^{\mathrm{kl}}(\bigodot_{i=1}^{t} \overset{\circ}{\exp}(-C\beta^{(i)})). \tag{9}$$

$\bigodot$ represents element-wise multiplication of a series of matrices. In particular, when each $\beta^{(i)}$ is fixed as $\Delta\beta$, the regularization $\beta$ satisfies $\beta = t\Delta\beta$, which yields the following explicit expression:

$$\hat{P}^{(t)} = \hat{\Gamma}_{\mathbb{U}}^{\mathrm{kl}}(\overset{\circ}{\exp}(-t\Delta\beta C)) = \hat{P}_\beta, \quad \beta = t\Delta\beta. \tag{10}$$

The $\hat{P}_\beta$ can admit the form $\hat{P}_\beta = D_{(\boldsymbol{u})} \overset{\circ}{\exp}(-\beta C) D_{(\boldsymbol{v})}$ and satisfies

$$\rho_{\mathbb{U}}(P_\beta^0) > \rho_{\mathbb{U}}(\hat{P}_\beta) > 0, \tag{11}$$

where $\rho_{\mathbb{U}}(P) = \|P\mathbf{1} - \boldsymbol{a}\|_1 + \|P^\top\mathbf{1} - \boldsymbol{b}\|_1$ represents the marginal error and $P_\beta^0 = D_{(\frac{1}{n}\mathbf{1})} \overset{\circ}{\exp}(-\beta C) D_{(\frac{1}{n}\mathbf{1})}$ is a classical initialization for KL-projections. The next Lemma establishes a theoretical bridge between the inexact transport plan and the exact transport plan.

**Lemma 2 (Inexact to Exact)** *Let $\hat{P}_\beta$ be an approximation to $P_\beta$,*

$$P_\beta = \Gamma_{\mathbb{U}}^{\text{kl}}(\hat{P}_\beta). \tag{12}$$

This result enables the inexact approximation $\hat{P}_\beta$ to serve as a warm start for computing the exact solution $P_\beta$ via the KL-projection.

Building on the above results, we propose an IP-EOT to provide an approximation $\hat{P}_\beta$, as outlined in Algorithm 1. The algorithm avoids the risk of numerical underflow by dividing the total regularization $\beta$ into $\ell$ steps (line 2) and progressively constructing the kernel matrix (line 6). Our matrices $\Delta K$ and $\hat{P}$ are much less prone to numerical underflow than the Sinkhorn kernel matrix $\overset{\circ}{\exp}(-\beta C)$. This advantage becomes more pronounced with a smaller $\beta$, leading to substantially improved numerical stability. With the aid of the IC preconditioner (line 4), line 5 employs a preconditioned conjugate gradient (PCG) method for computing $\Delta\boldsymbol{z}$. Since the cost of each iteration is $\mathcal{O}(n^2)$, the computational complexity of the algorithm is $\mathcal{O}(\ell n^2)$.

---

**Algorithm 1** IP-EOT

**Require:** Probability vectors $\{\boldsymbol{a}, \boldsymbol{b}\}$ with length $m$ and $n$, cost matrix $C$, entropic parameter $\beta$, proximal parameter $\Delta\hat{\beta}$
1: $\ell = \lceil \frac{\beta}{\Delta\hat{\beta}} \rceil$
2: $\Delta\beta \leftarrow \beta/\ell$
3: $\Delta K \leftarrow \overset{\circ}{\exp}(-\Delta\beta C)$
4: $\hat{P} \leftarrow \mathbf{1}_m\mathbf{1}_n^\top/(mn)$
5: **for** $t = 1, 2, \cdots, \ell$
6:     $K \leftarrow \Delta K \odot \hat{P}$
7:     $\boldsymbol{u} \leftarrow \boldsymbol{a} \oslash (K\boldsymbol{v}), \; \boldsymbol{v} \leftarrow \boldsymbol{b} \oslash (K^\top\boldsymbol{u})$
8:     $\hat{P} \leftarrow \text{diag}(\boldsymbol{u})K\text{diag}(\boldsymbol{v})$
9: **end**
10: **return** $\hat{P}$

---

**Algorithm 2** Fast Sparse Newton

**Require:** Initialized plan $P_0$, Probability vectors $\{\boldsymbol{a}, \boldsymbol{b}\}$, $C$, marginal error threshold $\rho_{th}$, $\beta$
1: **while** $\rho > \rho_{th}$ **do**
2:     Compute $H, \nabla f, \rho$ with (15)
3:     $\hat{H} \leftarrow \text{Sparsify}(H, \rho)$
4:     $L \leftarrow \text{IC}(\hat{H})$          $\triangleright \# \hat{H} \approx LL^\top$
5:     $\Delta\boldsymbol{z} \leftarrow \text{PCG}(\hat{H}, -\nabla f, L)$
6:     **if** $\alpha$ is searched **then** $\boldsymbol{z} \leftarrow \boldsymbol{z} + \alpha\Delta\boldsymbol{z}$
7:     **else** Update $\boldsymbol{z}$ by Sinkhorn scaling
8:     Compute marginal error $\rho$
9: **end while**
10: $(\boldsymbol{x}, \boldsymbol{y}) \leftarrow \boldsymbol{z}$
11: $P \leftarrow \overset{\circ}{\exp}\left(\beta(-C + \boldsymbol{x}\mathbf{1}^\top + \mathbf{1}\boldsymbol{y}^\top) - 1\right)$
12: **return** $P$

---

Since the iteration number of IP-EOT is tied to $\Delta\beta$, the algorithm cannot adaptively stop based on marginal error. Thus, it should be augmented with alternative solvers, switching to Sinkhorn or Newton methods if the IP-EOT solution yields insufficient accuracy.

## 4 FAST SPARSE NEWTON METHOD FOR ENTROPIC OPTIMAL TRANSPORT

The Newton method is based on the Lagrange function $\mathcal{L}$ associated with the entropic OT (1):

$$\mathcal{L}(P, \boldsymbol{x}, \boldsymbol{y}) := \frac{1}{\beta} P \cdot \overset{\circ}{\log} P + C \cdot P - \boldsymbol{x} \cdot (P\mathbf{1} - \boldsymbol{a}) - \boldsymbol{y} \cdot (P^\top\mathbf{1} - \boldsymbol{b}). \tag{13}$$

Using the primal-dual and strong duality theory (Nocedal and Wright, 1999), one only needs to minimize the function $f$ to get the solution of (1):

$$f(\boldsymbol{z}) = f(\boldsymbol{x}, \boldsymbol{y}) = -\min_{P \in \mathbb{U}} \mathcal{L}(P, \boldsymbol{x}, \boldsymbol{y}), \tag{14}$$

The explicit form of $f(\boldsymbol{z})$ follows by substituting $P = \overset{\circ}{\exp}\left(\beta(-C + \boldsymbol{x}\mathbf{1}^\top + \mathbf{1}\boldsymbol{y}^\top) - 1\right)$ into (13). The corresponding gradient and Hessian matrices of $f$ are

$$\nabla_{\boldsymbol{z}} f(\boldsymbol{z}) = (\nabla_{\boldsymbol{x}} f(\boldsymbol{x}, \boldsymbol{y}), \nabla_{\boldsymbol{y}} f(\boldsymbol{x}, \boldsymbol{y})) = \left(P\mathbf{1} - \boldsymbol{a}, P^\top\mathbf{1} - \boldsymbol{b}\right),$$

$$H(\boldsymbol{z}) = \nabla^2 f(\boldsymbol{z}) = \nabla^2 f(\boldsymbol{x}, \boldsymbol{y}) = \beta \begin{bmatrix} \text{diag}(P\mathbf{1}) & P \\ P^\top & \text{diag}(P^\top\mathbf{1}) \end{bmatrix}. \tag{15}$$

If $H(\boldsymbol{z}_k)$ is invertible, the Newton method updates $\boldsymbol{z}_{k+1}$ via the formula

$$\boldsymbol{z}_{k+1} = \boldsymbol{z}_k + \Delta\boldsymbol{z}_k, \quad \Delta\boldsymbol{z}_k = H(\boldsymbol{z}_k)^{-1}\left(-\nabla_{\boldsymbol{z}} f(\boldsymbol{z}_k)\right). \tag{16}$$

To avoid costly matrix inversion, $\Delta\boldsymbol{z}$ is obtained by solving a linear system $H\Delta\boldsymbol{z} = -\nabla f$, which is commonly solved by the Conjugate Gradient (CG) method (Tang et al., 2024). However, a key challenge arises from the $\mathcal{O}(n^3)$ computational cost per Newton iteration and the non-invertibility of the Hessian matrix $H$.

## 4.1 FAST POSITIVE-DEFINITE-PRESERVING SPARSE SCHEME

Prior research has made substantial progress in characterizing complexity and irreversibility by sparsifying the Hessian. Tang et al. (2024) observed that, after some Sinkhorn iterations, the Hessian matrix can be well-approximated by a sparse structure. Building on this observation, they reduced the computational complexity of the single Newton update (16) from $\mathcal{O}(n^3)$ to $\mathcal{O}(n^2)$ by enforcing a predetermined sparsity pattern on the Hessian matrix. Following this direction, Tang and Qiu (2024) proposed an improved sparsification strategy that ensures the invertibility of the sparsified Hessian matrix. However, both approaches suffer from a significant increase in computation time on certain large-scale problems. To overcome this limitation, we propose an efficient sparsification scheme that guarantees an invertible sparse approximation of the Hessian matrix $H$.

The sparsified matrix $\hat{H}$ is constructed by enforcing sparsity on the anti-diagonal submatrices $P$ and $P^\top$:

$$\hat{H} = \beta \begin{bmatrix} \operatorname{diag}(P\mathbf{1}) & \hat{P} \\ \hat{P}^\top & \operatorname{diag}(P^\top\mathbf{1}) \end{bmatrix}. \tag{17}$$

The sparsification procedure comprises the following two steps:

$$\widehat{P}^0_{ij} = \begin{cases} P_{ij}, & P_{ij} \geq \tau, \\ 0, & P_{ij} < \tau, \end{cases} \quad \widehat{P}_{ij} = \begin{cases} 0, & j = \arg\min_{k:\,\widehat{P}^0_{ik}>0} \widehat{P}^0_{ik}, \ (P - \widehat{P}^0)_{i\cdot} = \mathbf{0}, \\ 0, & i = \arg\min_{k:\,\widehat{P}^0_{kj}>0} \widehat{P}^0_{kj}, \ (P - \widehat{P}^0)_{\cdot j} = \mathbf{0}, \\ \widehat{P}^0_{ij}, & \text{otherwise}. \end{cases} \tag{18}$$

where $\tau = \|\nabla f\|_1$ is guided by the objective of achieving quadratic convergence (detailed in the proof of Theorem 2). The second step, diagonal dominating operation, ensures that all rows of $\hat{P}$ and $\hat{P}^\top$ contain truncated elements, which also implies that $\hat{H}$ exhibits diagonal dominance and positive definiteness. Appendix H presents the relationship between the sparsity level and $\beta$, and describes how to avoid excessive sparsification which may slow down the optimization.

Excessive sparsification, which removes important entries from the Hessian, may slow down the optimization. To avoid this, we reduce $\tau$ whenever the number of nonzeros in $\hat{P}$ falls below $2n$, ensuring that $\hat{P}$ retains at least $2n$ entries. The choice of $2n$ is motivated by the fact that the solution of the corresponding OT contains at most $2n - 1$ nonzero entries. Thus, if $\hat{P}$ contains more than $2n$ nonzeros, the corresponding $\hat{H}$ can be regarded as reliable. The detailed procedure can be found in the Appendix H.

**Theorem 2 (Local Quadratic Convergence)** *Let the sequence $\{\boldsymbol{z}_k\}$ be generated by*

$$\boldsymbol{z}_{k+1} = \boldsymbol{z}_k - \hat{H}_k^{-1}\nabla f(\boldsymbol{z}_k), \tag{19}$$

*where $\hat{H}_k$ is sparsified from the Hessian matrix $H_k = H(\boldsymbol{z}_k)$ by the fast positive-definite-preserving sparse scheme. The sequence $\{\boldsymbol{z}_k\}$ converges locally quadratically to the minimizer $\boldsymbol{z}^*$.*

## 4.2 PRECONDITIONER FOR NEWTON METHOD

To enhance the efficiency of each Newton iteration, we employ an incomplete Cholesky (IC) preconditioner to concentrate the eigenvalue distribution of the Hessian matrix, thereby reducing the number of CG iterations. This design leverages the fact that CG converges more rapidly when the eigenvalues of $H$ are tightly clustered (Hestenes et al., 1952) (see Appendix for details). The effectiveness of IC is demonstrated in Figure 9, which compares it with the diagonal preconditioner used in previous Newton-based algorithms for entropic OT (Brauer et al., 2017; Tang and Qiu, 2024).

### 4.3 Line Search with Sinkhorn Fallback

The step size is typically determined by a line search procedure, such as backtracking, to ensure that the objective value decreases at each Newton iteration. In a backtracking line search, the step size is initialized to $\alpha = 1$ and then repeatedly reduced by multiplying it with a constant factor $\in (0, 1)$ (e.g., 0.5) until $f(\boldsymbol{z}_k + \alpha \, \Delta \boldsymbol{z}_k) < f(\boldsymbol{z}_k)$. However, a practical issue arises when the line search yields an extremely small step size, causing the algorithm to enter a plateau phase in which the score barely decreases despite a considerable number of iterations. Tang and Qiu (2024) mitigated this issue by progressively strengthening the diagonal of the Hessian matrix, which shortens but does not completely eliminate the stagnation period.

To further address this problem, we propose a novel strategy. If the step size $\alpha$ obtained from a backtracking line search fails to produce an improvement in the objective value

$$f(\boldsymbol{z}_k + \alpha \Delta \boldsymbol{z}_k) > f(\boldsymbol{z}_k), \ \ \forall \alpha \in \{1, 0.5, 0.5^2, 0.5^3, 0.5^4\}, \tag{20}$$

then the algorithm employs the Sinkhorn scaling to refine the solution. This can generate a more robust direction $\boldsymbol{p}_k$ to escape the stagnation region and resume effective progress

$$f(\boldsymbol{z}_k + \boldsymbol{p}_k) < f(\boldsymbol{z}_k). \tag{21}$$

Once a suitable step size $\alpha$ is found, the method resumes Newton updates, achieving both robustness and rapid convergence.

The robustness of this first-order direction is underpinned by its satisfaction of the Armijo and Wolfe conditions, as formalized in the following lemma. This not only guarantees sufficient descent but also prevents continued stagnation, thereby re-establishing the conditions necessary for the subsequent Newton steps to resume their rapid convergence.

**Lemma 3** *Let $f(\boldsymbol{z})$ be defined in equation (14), and the sequence of search directions $\{\boldsymbol{p}_k\}$ be generated by Sinkhorn iterations. Then in each update, direction $\boldsymbol{p}_k$ with unit step size $\alpha_k = 1$ satisfies the following conditions:*

$$f(\boldsymbol{z}_k + \boldsymbol{p}_k) \leq f(\boldsymbol{z}_k) + c_1 \nabla f(\boldsymbol{z}_k)^\top \boldsymbol{p}_k, \tag{22}$$

$$\nabla f(\boldsymbol{z}_k + \boldsymbol{p}_k)^\top \boldsymbol{p}_k \geq c_2 \nabla f(\boldsymbol{z}_k)^\top \boldsymbol{p}_k, \tag{23}$$

*for constants $0 < c_1 < c_2 < 1$.*

### 4.4 Fast Sparse Newton Method

Algorithm 2 outlines the fast sparse Newton method for entropic OT. In each iteration, the Hessian is sparsified (line 3) and preconditioned using an incomplete Cholesky (IC) factorization $L$ (line 4). The main computational cost lies in computing $\Delta \boldsymbol{z}$, which requires $\mathcal{O}(n\|\hat{H}\|_0)$ operations, where $\|\hat{H}\|_0$ denotes the number of nonzero elements (line 5). At line 6, the step size $\alpha$ is selected via a backtracking line search over $\{0.5, 0.5^2, 0.5^3, 0.5^4\}$. If none of these values improves the objective, we apply the Sinkhorn scaling for refinement. This adjustment helps Newton's method to escape flat regions, where progress would otherwise stall due to vanishing step sizes, and is critical for establishing global convergence (line 7).

**Theorem 3 (Global Convergence)** *The sequence $\{\boldsymbol{z}_k\}$ generated by Algorithm 2 converges globally to the minimizer $\boldsymbol{z}^*$.*

## 5 Proximal-Sinkhorn-Newton Method

This section presents the main algorithm, the Proximal-Sinkhorn-Newton (PSN), in Algorithm 3. The algorithm begins with the IP-EOT to produce an approximate solution $\hat{P}_\beta$. If $\hat{P}_\beta$ meets the predefined criteria, it is returned as the final output. Otherwise, the algorithm refines the solution by switching to either the Sinkhorn scaling or the fast sparse Newton method, depending on the sparsity of $\hat{H}$. If $\hat{H}$ contains fewer than $\lambda n$ non-zero elements, the fast sparse Newton method is employed; otherwise, the Sinkhorn scaling is used. This approach is motivated by the relationship between sparsity and computational efficiency: Newton's method benefits from sparser matrices, which occur as $\beta$ increases (Tang et al., 2024). Thus, the strategy automatically favors Newton steps for larger $\beta$ and Sinkhorn scaling for smaller $\beta$.

**Theorem 4** *Algorithm 3 converges globally to the solution of the corresponding entropic OT.*

---

**Algorithm 3** Proximal-Sinkhorn-Newton (PSN)

---

**Require:** Probability vectors $\{\boldsymbol{a}, \boldsymbol{b}\}$, cost matrix $C$, entropic parameter $\beta$, proximal parameter $\Delta\hat{\beta}$, marginal error threshold $\rho_{th}$, target sparsity $\lambda$.
**Ensure:** The transport plan $P$ for entropic OT with parameter $\beta$

1: $\hat{P}_\beta, u, v \leftarrow$ IP-EOT$(\boldsymbol{a}, \boldsymbol{b}, C, \beta, \Delta\hat{\beta})$          $\triangleright$ `# IP-EOT for approximation`
2: $\rho = \left\| \hat{P}_\beta \mathbf{1} - \boldsymbol{a} \right\|_1 + \left\| \hat{P}_\beta^\top \mathbf{1} - \boldsymbol{b} \right\|_1$
3: **if** $\rho < \rho_{th}$ **then** $P = \hat{P}_\beta$
4: **else**
5:     $\hat{H} \leftarrow$ Sparsify Hessian$(\hat{P}_\beta, \rho)$ by (15) and Sec. 4.1
6:     **if** $\|\hat{H}\|_0 < \lambda n$ **then**
7:        $P, \rho \leftarrow$ Fast Sparse Newton $\left( \boldsymbol{a}, \boldsymbol{b}, C, \hat{P}_\beta, \rho_{th}, \beta \right)$
8:     **else**
9:        **while** $\rho > \rho_{th}$ **do**
10:           $\boldsymbol{u} \leftarrow \boldsymbol{a} \oslash (\hat{P}_\beta \boldsymbol{v}), \boldsymbol{v} \leftarrow \boldsymbol{b} \oslash (\hat{P}_\beta^\top \boldsymbol{u})$       $\triangleright$ `# Sinkhorn scaling`
11:           Compute $\rho$ with $P = \text{diag}(\boldsymbol{v})\hat{P}_\beta \text{diag}(\boldsymbol{u})$
12:        **end while**
13:     **end if**
14: **end if**
15: **return** $P$

---

## 6 NUMERICAL EXPERIMENTS

We evaluate the proposed algorithm PSN and other contributions from the following three parts:

       Q1. How does the performance of IP-EOT compare to the Sinkhorn algorithm?

       Q2. What are the advantages of using an IC preconditioner in the Newton method?

       Q3. How does PSN perform across different problem sizes and regularization strengths?

**Datasets.** We consider three datasets following the experimental setup in (Tang et al., 2024): (1) Random linear assignment: $n \in \{500, 4000\}$, the cost matrix $C \in \mathbb{R}^{n \times n}$ where are entries sampled from a uniform distribution over $[0, 1]$; uniform marginals $\boldsymbol{a} = \boldsymbol{b} = \frac{1}{500}\mathbf{1}$. This is a fundamental problem in graph matching. (2) MNIST: each image is flattened and normalized to form marginals; the cost matrix is defined by the squared Euclidean distance $\|x - y\|_2$; $n \in \{784, 4096\}$; (3) Multiple-solutions variant: based on MNIST, the cost matrix replaced with $\ell_1$ norm $\|x - y\|_1$ to remove uniqueness in the OT solution, which is a challenge for Newton-based algorithms (Tang et al., 2024). Finally, we set $\beta = 200 \ln n$ for weak regularization, as the distance between $P_\beta$ and the optimal plan $P_\infty$ scales with $\frac{\ln n}{\beta}$ (Altschuler et al., 2017). Under this setting, the values of $\beta$ in experiments at the hundred-scale align with those reported in previous studies, making the comparison appropriate.

**Setting.** In PSN, the proximal parameter $\Delta\hat{\beta}$ is set to 25 for general OT. This ensures numerical stability and leverages the rapid early convergence of IP-EOT, as the iteration count is $\beta/\Delta\hat{\beta}$. Given the faster convergence of the Sinkhorn method in linear assignment (Altschuler et al., 2017), we set $\Delta\hat{\beta} = 50$ for this special case. The sparsity parameter $\lambda$ is set to 30 for linear assignment and 70 for general OT. This design is informed by the theoretical sparsity of OT solutions ($n$ for linear assignment and $2n - 1$ for general OT) (Peyré et al., 2017) and the fact that entropic OT solutions are typically denser (Tang et al., 2024).

**Baselines** include the Sparse-Newton-Sinkhorn method (SNS, ICLR) (Tang et al., 2024), the Safe and Sparse Newton Method[1] (SSNS, NeurIPS) (Tang and Qiu, 2024), the Newton method for entropic OT[2] (Brauer et al., 2017), the log-Sinkhorn (Benamou et al., 2015), and the primal Sinkhorn (Cuturi, 2013) (when $\beta > 100$, the primal Sinkhorn may exhibit numerical instability, as reported in Appendix

---

[1]https://github.com/yixuan/regot-python
[2]https://github.com/dirloren/sinkhornnewton

F of (Shen et al., 2024)). All methods were benchmarked in terms of runtime using MATLAB R2024b on an Intel Core i7-12800HX processor[3].

## 6.1 IP-EOT vs. log-Sinkhorn

This section compares IP-EOT with log-Sinkhorn. Figure 3 shows that the error of IP-EOT decreases much faster than that of log-Sinkhorn in the early iterations. Moreover, since the per-iteration cost of IP-EOT is less than that of log-Sinkhorn, its efficiency advantage in the early iterations becomes even more pronounced.

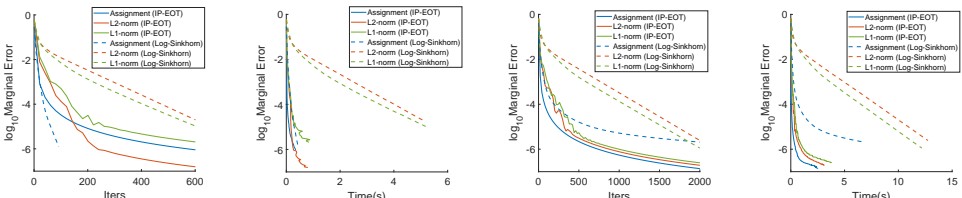

Figure 3: IP-EOT vs. log-Sinkhorn: $\beta = 500$ for the first two subfigures, $\beta = 1200$ for the latter two.

## 6.2 COMPARISON BETWEEN INCOMPLETE CHOLESKY AND DIAGONAL PRECONDITIONER

This section compares the improvements provided by the Incomplete Cholesky (IC) and diagonal preconditioners when applied to the Newton method. Figure 4 shows that the IC preconditioner enhances spectral clustering, significantly reduces CG iterations, and consistently outperforms the diagonal preconditioner across all datasets. Since Theorem 5 shows that CG's per-step error is mainly governed by the clustered bulk of eigenvalues, these few isolated eigenvalues do not noticeably reduce the performance gains.

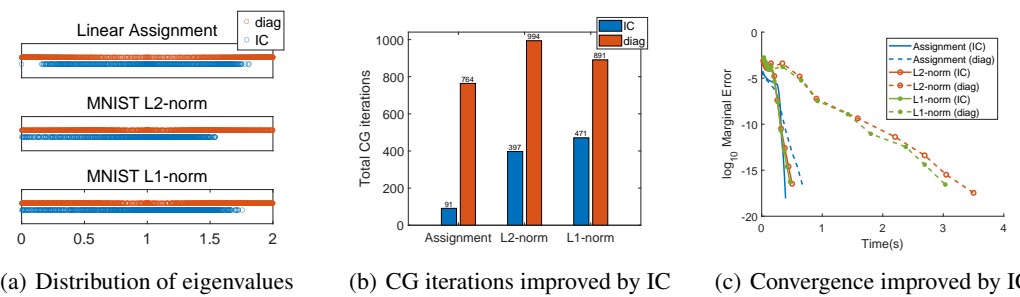

(a) Distribution of eigenvalues   (b) CG iterations improved by IC   (c) Convergence improved by IC

Figure 4: Comparison with diagonal and IC preconditioners. 300 Sinkhorn iterations are used to generate initial approximations.

## 6.3 COMPARISON BETWEEN PSN AND OTHER ALGORITHMS

This section evaluates PSN against several baseline methods. As illustrated in Figure 5, PSN demonstrates strong performance across all settings. In (a)–(c), PSN circumvents the numerical overflow risk inherent in the Sinkhorn method through a few proximal steps, achieving comparable efficiency safely and surpassing all other baselines. Subfigures (d)–(f) show that for large $\beta$, PSN outperforms other algorithms, especially log-Sinkhorn. In larger-scale problems (g)–(i), the Newton-based baselines suffer a sharp runtime increase, whereas PSN remains efficient and achieves over $20\times$ speedup over SNS and SSNS in some cases. Further results on iterations and time to accuracy are included in the Appendix.

---

[3]For SSNS implemented in C++, its runtime was scaled to the MATLAB baseline by comparing the execution time of the Newton method in C++ and MATLAB.

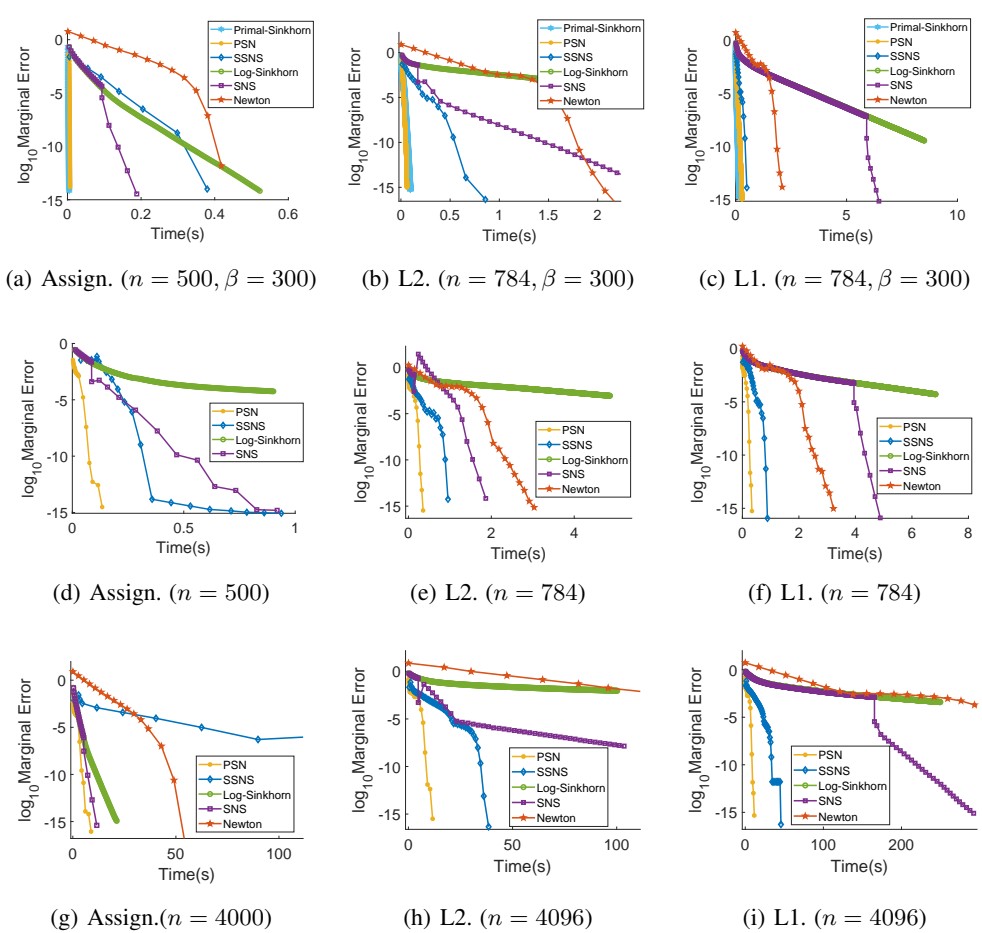

Figure 5: Comparisons between PSN and other algorithms on different problem sizes and different $\beta$. For subfigures without an explicit $\beta$ value, the default setting $\beta = 200 \ln(n)$ is adopted.

## 7 CONCLUSION

We elaborate the trajectory of the regularization parameter during the iterations of the proximal point method and establish its ability to approximate the entropic optimal transport solution. This theoretical insight facilitates the seamless integration of the proximal point method with complementary algorithms, leading to the proposed Proximal-Sinkhorn-Newton framework that achieves high efficiency and robustness under varying entropic regularization parameters. Moreover, compared with previous Newton-based algorithms, the proposed fast Sparse Newton method accelerates each iteration with a preconditioner and mitigates stagnation during optimization through a novel line search strategy that incorporates Sinkhorn scaling. Experimental results show that PSN significantly faster than the state-of-the-art methods.

Since the number of IP-EOT iterations in this work is chosen empirically, future research may investigate how the number of proximal point iterations influences overall convergence, potentially leading to more adaptive iteration scheduling strategies. Another promising direction is the integration of additional algorithmic modules into the proximal point framework to further enhance performance. From a practical standpoint, exploiting hardware-based acceleration also holds considerable potential for improving computational efficiency.

**Ethics Statement.** This work focuses on the development and analysis of algorithms for entropic optimal transport. It does not involve human subjects, sensitive data, or applications with foreseeable ethical risks. To the best of our knowledge, this research raises no ethical concerns.

**Reproducibility Statement.** To ensure reproducibility, we provide the source code with our submission. All algorithmic steps, parameter settings, and experimental configurations are explicitly described in the paper. With the released code and detailed specifications, independent researchers should be able to fully reproduce our results.

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

## A  NOTATIONS

Key notations are summarized below.

Table 1: Symbols and Notations.

| Symbol | Definition |
|---|---|
| $\mathcal{H}(\cdot)$ | entropy function |
| $\mathcal{B}(\cdot)$ | Bregman divergence |
| $\mathbb{U}$ | $\left\{P \in \mathbb{R}_+^{n \times n} : P\mathbf{1} = \boldsymbol{a}, P^\top\mathbf{1} = \boldsymbol{b}\right\}, \boldsymbol{a}, \boldsymbol{b} \in \mathbb{R}^n$ |
| $D_{(\boldsymbol{x})}$ | diagonal matrix of a vector $\boldsymbol{x}$ |
| $A \cdot B$ | Frobenius inner product between matrix $A$ and $B$ |
| $\otimes$ | Kronecker product |
| $\overset{\circ}{\exp}$ | element-wise exponential |
| $\oslash$ | element-wise division |
| $\odot$ | element-wise product |
| $\bigodot$ | element-wise multiplication of a series of matrices |
| $\beta$ | the parameter in entropic OT |
| $\Delta\beta$ | the proximal parameter in proximal point methods |
| $\Gamma_{\mathbb{U}}^{\mathrm{kl}}(\cdot), \hat{\Gamma}_{\mathrm{kl}}(\cdot)$ | KL-projection and its approximating version |
| $P_\beta, \hat{P}_\beta$ | the exact and approximating solution of OT with entropic parameter $\beta$ |

## B  PROOFS FOR PROXIMAL POINT METHODS

**Lemma 1 (Explicit form of $\boldsymbol{\beta}$)** *The iterative schemes of the exact/inexact proximal point method for entropic OT admit the following explicit representation:*

$$P^{(t)} = \Gamma_{\mathbb{U}}^{\mathrm{kl}}(\bigodot_{i=1}^{t} \overset{\circ}{\exp}(-C\beta^{(i)})), \quad \hat{P}^{(t)} = \hat{\Gamma}_{\mathbb{U}}^{\mathrm{kl}}(\bigodot_{i=1}^{t} \overset{\circ}{\exp}(-C\beta^{(i)})). \tag{24}$$

**Proof** *This proof begins with the iterative formula of the exact proximal point method:*

$$P^{(t)} = \Gamma_{\mathbb{U}}^{\mathrm{kl}} \left( \overset{\circ}{\exp}(-C\beta^{(t)}) \odot P^{(t-1)} \right). \tag{25}$$

*For the first step, we have*

$$P^{(1)} = \Gamma_{\mathbb{U}}^{\mathrm{kl}}(\overset{\circ}{\exp}(-C\beta^{(1)})) = D_{(\boldsymbol{u}_1)} \overset{\circ}{\exp}(-\beta^{(1)}C) \, D_{(\boldsymbol{v}_1)}, \tag{26}$$

*An equivalent form (Shen et al., 2024, Proposition 3) can be derived via elementary matrix operations as*

$$P^{(1)} = \overset{\circ}{\exp}\left(-C\beta^{(1)}\right) \odot (\boldsymbol{u}_1 \otimes \boldsymbol{v}_1^\top). \tag{27}$$

*Hence, for the subsequent iteration, we have:*

$$P^{(2)} = \Gamma_{\mathbb{U}}^{\mathrm{kl}} \left( \overset{\circ}{\exp}(-C\beta^{(2)}) \odot P^{(1)} \right) \tag{28}$$

$$= \Gamma_{\mathbb{U}}^{\mathrm{kl}} \left( \overset{\circ}{\exp}(-C\beta^{(2)}) \odot \overset{\circ}{\exp}(-C\beta^{(1)}) \odot (\boldsymbol{u}_1 \otimes \boldsymbol{v}_1^\top) \right) \tag{29}$$

$$= \left( \overset{\circ}{\exp}(-C\beta^{(2)}) \odot \overset{\circ}{\exp}(-C\beta^{(1)}) \odot (\boldsymbol{u}_1 \otimes \boldsymbol{v}_1^\top) \right) \odot (\boldsymbol{u}_2 \otimes \boldsymbol{v}_2^\top) \tag{30}$$

$$= \overset{\circ}{\exp}(-C(\beta^{(2)} + \beta^{(1)})) \odot (\boldsymbol{u}_1 \otimes \boldsymbol{v}_1^\top) \odot (\boldsymbol{u}_2 \otimes \boldsymbol{v}_2^\top) \tag{31}$$

$$= \overset{\circ}{\exp}(-C(\beta^{(2)} + \beta^{(1)})) \odot \left( (\boldsymbol{u}_1 \odot \boldsymbol{u}_2) \otimes (\boldsymbol{v}_1^\top \odot \boldsymbol{v}_2^\top) \right). \tag{32}$$

*By defining $\boldsymbol{u}_3 := \boldsymbol{u}_1 \odot \boldsymbol{u}_2$ and $\boldsymbol{v}_3 := \boldsymbol{v}_1 \odot \boldsymbol{v}_2$, we have*

$$P^{(2)} = \mathring{\exp}(-C(\beta^{(2)} + \beta^{(1)})) \odot (\boldsymbol{u}_3 \otimes \boldsymbol{v}_3^\top). \tag{33}$$

*Due to $P^{(2)} \in \mathbb{U}$, $P^{(2)}$ is the KL-projection of the kernel matrix $\mathring{\exp}\big(-C(\beta^{(1)} + \beta^{(2)})\big)$. Since the KL-projection is unique for a given kernel matrix (Cuturi, 2013), it follows that*

$$P^{(2)} = \Gamma_{\mathbb{U}}^{\mathrm{kl}}(\bigodot_{i=1}^{2} \mathring{\exp}(-C\beta^{(i)})). \tag{34}$$

*The same structure holds for general $t$ by induction:*

$$P^{(t)} = \Gamma_{\mathbb{U}}^{\mathrm{kl}}(\bigodot_{i=1}^{t} \mathring{\exp}(-C\beta^{(i)})). \tag{35}$$

*For the inexact proximal point method, we can also obtain*

$$\hat{P}^{(2)} = \hat{\Gamma}_{\mathbb{U}}^{\mathrm{kl}}\left(\mathring{\exp}(-C\beta^{(2)}) \odot \hat{P}^{(1)}\right) \tag{36}$$

$$= \hat{\Gamma}_{\mathbb{U}}^{\mathrm{kl}}\left(\mathring{\exp}(-C\beta^{(2)}) \odot \mathring{\exp}(-C\beta^{(1)}) \odot (\hat{\boldsymbol{u}}_1 \otimes \hat{\boldsymbol{v}}_1^\top)\right) \tag{37}$$

$$= \left(\mathring{\exp}(-C\beta^{(2)}) \odot \mathring{\exp}(-C\beta^{(1)}) \odot (\hat{\boldsymbol{u}}_1 \otimes \hat{\boldsymbol{v}}_1^\top)\right) \odot (\hat{\boldsymbol{u}}_2 \otimes \hat{\boldsymbol{v}}_2^\top) \tag{38}$$

$$= \mathring{\exp}(-C(\beta^{(2)} + \beta^{(1)})) \odot (\hat{\boldsymbol{u}}_1 \otimes \hat{\boldsymbol{v}}_1^\top) \odot (\hat{\boldsymbol{u}}_2 \otimes \hat{\boldsymbol{v}}_2^\top) \tag{39}$$

$$= \mathring{\exp}(-C(\beta^{(2)} + \beta^{(1)})) \odot (\underbrace{(\hat{\boldsymbol{u}}_1 \odot \hat{\boldsymbol{u}}_2)}_{\hat{\boldsymbol{u}}_3} \otimes \underbrace{(\hat{\boldsymbol{v}}_1^\top \odot \hat{\boldsymbol{v}}_2^\top)}_{\hat{\boldsymbol{v}}_3^\top}) \tag{40}$$

$$= \mathring{\exp}(-C(\beta^{(2)} + \beta^{(1)})) \odot (\hat{\boldsymbol{u}}_3 \otimes \hat{\boldsymbol{v}}_3^\top). \tag{41}$$

*Since $\hat{\boldsymbol{u}}_3$ and $\hat{\boldsymbol{v}}_3$ are approximations for the scaling vectors of the corresponding kernel matrix $\mathring{\exp}(-C(\beta^{(2)} + \beta^{(1)}))$, $\hat{P}^{(2)}$ can be rewritten as*

$$\hat{P}^{(2)} = \hat{\Gamma}_{\mathbb{U}}^{\mathrm{kl}}(\bigodot_{i=1}^{2} \mathring{\exp}(-C\beta^{(i)})). \tag{42}$$

*The same structure holds for general $t$ by induction:*

$$\hat{P}^{(t)} = \hat{\Gamma}_{\mathbb{U}}^{\mathrm{kl}}(\bigodot_{i=1}^{t} \mathring{\exp}(-C\beta^{(i)})). \tag{43}$$

**Lemma 2 (Inexact to Exact)** *Let $\hat{P}_\beta$ be an approximation to $P_\beta$,*

$$P_\beta = \Gamma_{\mathbb{U}}^{\mathrm{kl}}(\hat{P}_\beta). \tag{44}$$

**Proof** *Since $\hat{P}_\beta$ is a approximation of $P_\beta$, we have*

$$\hat{P}_\beta = D_{(\hat{\boldsymbol{u}})} \, \mathring{\exp}(-\beta C) \, D_{(\hat{\boldsymbol{v}})}, \tag{45}$$

*which can be rewritten as*

$$\hat{P}_\beta = \mathring{\exp}(-\beta C) \odot (\hat{\boldsymbol{u}} \otimes \hat{\boldsymbol{v}}^\top), \tag{46}$$

*according to (Shen et al., 2024, Proposition 3). Applying the KL-projection yields*

$$\Gamma_{\mathbb{U}}^{\mathrm{kl}}(\hat{P}_\beta) = \mathring{\exp}(-\beta C) \odot (\hat{\boldsymbol{u}} \otimes \hat{\boldsymbol{v}}^\top) \odot (\boldsymbol{u}_2 \otimes \boldsymbol{v}_2^\top) \tag{47}$$

$$= \mathring{\exp}(-\beta C) \odot (\underbrace{(\hat{\boldsymbol{u}} \odot \boldsymbol{u}_1)}_{\boldsymbol{u}_2} \otimes \underbrace{(\hat{\boldsymbol{v}}_1^\top \odot \hat{\boldsymbol{v}}^\top)}_{\boldsymbol{v}_2^\top}) \tag{48}$$

$$= \mathring{\exp}(-\beta C) \odot (\boldsymbol{u}_2 \otimes \boldsymbol{v}_2^\top) \tag{49}$$

*Since the KL-projection is unique for a given kernel matrix (Cuturi, 2013), it follows that $\Gamma_{\mathbb{U}}^{\mathrm{kl}}(\hat{P}_\beta) = \Gamma_{\mathbb{U}}^{\mathrm{kl}}(\mathring{\exp}(-\beta C)) = P_\beta$.*

**Theorem 1** *Inexact-to-Exact Transition*
*If $\hat{P}^{(t)}$ denotes the transport plan updated at the $t$-th iteration in the inexact proximal point method with a fixed proximal parameter $\Delta\beta$ , then*

$$P_\beta = \Gamma_{\mathbb{U}}^{\mathrm{kl}}(\hat{P}^{(t)}), \ where \ \beta = t\Delta\beta. \tag{50}$$

**Proof** *According to Lemma 1, we have*

$$\hat{P}^{(t)} = \hat{\Gamma}_{\mathrm{kl}}(\overset{\circ}{\exp}(-t\Delta\beta C)) = \hat{P}_\beta. \tag{51}$$

*Then, by Lemma 2, it follows that*

$$\Gamma_{\mathbb{U}}^{\mathrm{kl}}(\hat{P}_\beta) = P_\beta, \tag{52}$$

*which completes the proof.*

## C    PROOF FOR LOCAL QUADRATIC CONVERGENCE

**Theorem 2 (Local Quadratic Convergence)** *Let the sequence $\{z_k\}$ be generated by*

$$z_{k+1} = z_k - \hat{H}_k^{-1}\nabla f(z_k), \tag{53}$$

*where $\hat{H}_k$ is sparsified from the Hessian matrix $H_k = H(z_k)$ by the Fast-Definite-Positive Sparsify Scheme. The sequence $\{z_k\}$ converges locally quadratically to the minimizer $z^*$.*

**Proof** *Define that $\|x\| = \sum_i |x_i|$ if $x$ is a vector and $\|X\| = \sum_i\sum_j |x_{ij}|$ if $X$ is a matrix.*

*To establish quadratic convergence, we must show that there exists a constant $c > 0$ such that for sufficiently large $k$, the following inequality holds:*

$$\|z_{k+1} - z^*\| \le c\|z_k - z^*\|^2. \tag{54}$$

*We begin the convergence analysis by examining the error at iteration $k+1$ and applying the triangle inequality:*

$$
\begin{aligned}
\|z_{k+1} - z^*\| &= \|z_k + \Delta z - z^*\| \\
&= \left\|z_k - \hat{H}_k^{-1}\nabla f(z_k) - z^*\right\| \\
&\le \left\|\hat{H}_k^{-1}\right\|\left\|\hat{H}_k(z_k - z^*) - \nabla f(z_k)\right\| \\
&\le \left\|\hat{H}_k^{-1}\right\|\left\|\nabla f(z_k) - \nabla f(z^*) - H(z_k)(z_k - z^*) + H(z_k)(z_k - z^*) - \hat{H}_k(z_k - z^*)\right\| \\
&\le \underbrace{\left\|\hat{H}_k^{-1}\right\|}_{(I)}\left(\underbrace{\|\nabla f(z_k) - \nabla f(z^*) - H(z_k)(z_k - z^*)\|}_{(II)} + \underbrace{\left\|H(z_k) - \hat{H}_k\right\|}_{(III)}\|z_k - z^*\|\right).
\end{aligned}
\tag{55}
$$

*If we can prove that exist positive constants $c_1$, $c_2$, $c_3$, such that*

$$\|\hat{H}_k^{-1}\| \le c_1, \tag{56}$$

$$\|\nabla f(z_k) - \nabla f(z^*) - H(z_k)(z_k - z^*)\| \le c_2\|z_k - z^*\|^2, \tag{57}$$

$$\left\|H(z_k) - \hat{H}_k\right\| \le c_3\|z_k - z^*\|, \tag{58}$$

*then we get*

$$\|z_{k+1} - z^*\| \le c_1(c_2 + c_3)\|z_k - z^*\|^2, \tag{59}$$

*which implies quadratic convergence of $\{z_k\}$.*

*For the first part (I), we know that for symmetric matrix $\hat{H}_k$, $\|\hat{H}_k^{-1}\|_2 = 1/\lambda_{min}(\hat{H}_k)$. Therefore, $\exists\, a > 0$, we have*

$$\|\hat{H}_k^{-1}\| \le \frac{a}{\lambda_{min}(\hat{H}_k)}. \tag{60}$$

*Now we use Gershgorin's Circle Theorem to estimate the lower bound of $\lambda_{min}(\hat{H}_k)$. Define $\delta_i = \left|\hat{h}_{ii}\right| - \sum_{j \neq i}\left|\hat{h}_{ij}\right|$. Since the sparsification scheme eliminates the smallest entry at each step while preserving the diagonal elements, there exists $r > 0$ such that*

$$r < \delta_i = |h_{ii}| - \sum_{j \neq i}\left|\hat{h}_{ij}\right| = \left(\sum_{j \neq i}|h_{ij}|\right) - \left(\sum_{j \neq i}\left|\hat{h}_{ij}\right|\right) = \sum_{j \neq i}\left(|h_{ij}| - \left|\hat{h}_{ij}\right|\right). \tag{61}$$

*Therefore, it follows directly from Gershgorin's Circle Theorem that*

$$\lambda_{min}(\hat{H}_k) \geq \min_i \delta_i. \tag{62}$$

*Let $c_1 = a/r$, then we get the inequality*

$$\|\hat{H}_k^{-1}\| \leq c_1. \tag{63}$$

*For the second part (II), we define $U(\boldsymbol{z}^*)$ is the neighborhood of $\boldsymbol{z}^*$ and use the fact that $\nabla f$ is continuously differentiable on , and $H(\boldsymbol{z})$ is Lipschitz continuous on the same neighborhood. This Lipschitz continuity implies that there exists a constant $l$ such that*

$$\|H(\boldsymbol{z}_x) - H(\boldsymbol{z}_y)\| \leq l\|\boldsymbol{z}_x - \boldsymbol{z}_y\|, \quad \forall \boldsymbol{z}_x, \boldsymbol{z}_y \in U(\boldsymbol{z}^*). \tag{64}$$

*Based on these properties, an application of Theorem 3.2.12 in [ortega2000iterative] shows that for any $\boldsymbol{z}_k \in U(\boldsymbol{z}^*)$, the following inequality holds:*

$$\|\nabla f(\boldsymbol{z}_k) - \nabla f(\boldsymbol{z}^*) - H(\boldsymbol{z}_k)(\boldsymbol{z}^* - \boldsymbol{z}_k)\| \leq \frac{l}{2}\|\boldsymbol{z}_k - \boldsymbol{z}^*\|^2. \tag{65}$$

*Therefore, set $c_2 = l/2$ and we get*

$$\|\nabla f(\boldsymbol{z}_k) - \nabla f(\boldsymbol{z}^*) - H(\boldsymbol{z}_k)(\boldsymbol{z}^* - \boldsymbol{z}_k)\| \leq c_2\|\boldsymbol{z}_k - \boldsymbol{z}^*\|^2. \tag{66}$$

*For the third part (III), we bound the error of the Hessian approximation, $\|H_k - \hat{H}_k\|$. Let $E_k = H_k - \hat{H}_k$, and let $e_{ij}^{(k)}$ denote an entry of $E_k$. We divide $e_{ij}^{(k)}$ into two parts: one corresponding to the finite set $\mathcal{S}$ of index pairs $(i,j)$ where $h_{ij}^* = 0$, and the other corresponding to the set $\overline{\mathcal{S}}$, which contains index pairs where $h_{ij}^* \neq 0$. Therefore, we have*

$$\|H_k - \hat{H}_k\| = \sum_i \sum_j \left|e_{ij}^{(k)}\right| = \sum_{(i,j) \in \mathcal{S}}\left|e_{ij}^{(k)}\right| + \sum_{(i,j) \in \overline{\mathcal{S}}}\left|e_{ij}^{(k)}\right|. \tag{67}$$

*When $(i,j) \in \mathcal{S}$, we have $h_{ij}^* = 0$. We use Lipschiz continuity of $H(\boldsymbol{z})$ to bound the term $\left|e_{ij}^{(k)}\right|$,*

$$\left|e_{ij}^{(k)}\right| = \left|h_{ij}^{(k)} - h_{ij}^*\right| \leq \|H_k - H(\boldsymbol{z}^*)\| \leq l\|\boldsymbol{z}_k - \boldsymbol{z}^*\|. \tag{68}$$

*Let the size of $\mathcal{S}$ be $d$. We can bound the norm of $E_k$ by defining $c_3 = dl > 0$,*

$$\sum_{(i,j) \in \mathcal{S}}\left|e_{ij}^{(k)}\right| \leq d \cdot l\|\boldsymbol{z}_k - \boldsymbol{z}^*\| = c_3\|\boldsymbol{z}_k - \boldsymbol{z}^*\|. \tag{69}$$

*For the second part of (67), we have*

$$\sum_{(i,j) \in \overline{\mathcal{S}}}\left|e_{ij}^{(k)}\right| \leq \sum_{(i,j) \in \overline{\mathcal{S}}} \tau = c_4\tau. \tag{70}$$

*where $c_4 = 2n^2 - d$. Combining (69) and (70) we get that*

$$\sum_i \sum_j \left|e_{ij}^{(k)}\right| \leq c_3\|\boldsymbol{z}_k - \boldsymbol{z}^*\| + c_4\tau. \tag{71}$$

*Since $\tau = \|z_k - z^*\|$, it follows that*

$$\sum_i \sum_j \left| e_{ij}^{(k)} \right| \leq (c_3 + c_4) \|z_k - z^*\|, \tag{72}$$

*so we have*

$$\|H_k - \hat{H}_k\| = \sum_i \sum_j \left| e_{ij}^{(k)} \right| \leq (c_3 + c_4)\|z_k - z^*\|. \tag{73}$$

*Combining (63), (66), (73) and the formula of $\|z_{k+1} - z^*\|$, we obtain*

$$\|z_{k+1} - z^*\| \leq c_1 (c_2 + c_3 + c_4) \|z_k - z^*\|^2, \tag{74}$$

*implying the local quadratic convergence.*

## D    PROOF FOR GLOBAL CONVERGENCE

**Lemma 3** *Let $f(z)$ be defined in equation (14), and the sequence of search directions $\{p_k\}$ be generated by the Sinkhorn iterations. Then in each update, direction $p_k$ with unit step size $\alpha_k = 1$ satisfies the following conditions:*

$$f(z_k + p_k) \leq f(z_k) + c_1 \nabla f(z_k)^\top p_k, \tag{75}$$

$$\nabla f(z_k + p_k)^\top p_k \geq c_2 \nabla f(z_k)^\top p_k, \tag{76}$$

*for constants $0 < c_1 < c_2 < 1$.*

**Proof** *Since Sinkhorn algorithm is a Block Coordinate Descent method, equation (75) is naturally satisfied. Let us analyze a single step of updating the $y$ block while keeping the $x$ block fixed, and symmetric argument applies to the update of the $x$ block.*

*A Sinkhorn iteration updates the $y$ block by solving the subproblem:*

$$y_{k+1} = \arg\min_{y \in \mathbb{R}^n} f(x_{k+1}, y). \tag{77}$$

*The first-order optimality condition for this subproblem requires the partial gradient with respect to $y$ is zero, i.e.,*

$$\nabla_y f(x_{k+1}, y_{k+1}) = 0. \tag{78}$$

*The search direction $p_k$ associated with this update has a block structure $p_k = (0, p_k^y)$, as it only modifies the $y$ components. Consequently, the projection of the new gradient $\nabla f(z_{k+1})$ onto the search direction $p_k$ is:*

$$\nabla f(z_{k+1})^\top p_k = \nabla_x f(z_{k+1})^\top \cdot 0 + \nabla_y f(z_{k+1})^\top p_k^y = 0. \tag{79}$$

*Since the Sinkhorn iteration employs a descent direction, we have $\nabla f(z_k)^\top p_k < 0$. Combining with equation (79) it leads to*

$$c_2 \nabla f(z_k)^\top p_k < 0 = \nabla f(z_{k+1})^\top p_k, \tag{80}$$

*which satisfies the equation (76) for any $c_2 \in (c_1, 1)$. Thus, the conditions of this Lemma are always met by Sinkhorn iterations.*

**Lemma 4** *Let $f(z)$ be defined in equation (14), the sequences $\{z_k\}$ and $\{\alpha_k\}$ be generated by Algorithm 2, and $p_k$ be the search direction. Then, the function $f(z)$ and the generated sequences possess the following properties:*

- *$f(z)$ is bounded below, i.e., there exists a scalar $\bar{f}$ such that $f(z) \geq \bar{f}$ for all $z$.*

- *$f(z)$ is continuously differentiable.*

- *The gradient of $f$, $\nabla f$, is globally Lipschitz continuous, i.e., there exists a constant $l > 0$ such that*
$$\|\nabla f(z_x) - \nabla f(z_y)\| \leq l\|z_x - z_y\|, \quad \forall z_x, z_y. \tag{81}$$

- *The step size $\alpha_k$ is determined by a line search procedure that ensures the Wolfe conditions are satisfied for all $k$:*

$$f(\boldsymbol{z}_k + \alpha_k p_k) \leq f(\boldsymbol{z}_k) + c_1 \alpha_k \nabla f(\boldsymbol{z}_k)^\top p_k, \tag{82}$$

$$\nabla f(\boldsymbol{z}_k + \alpha_k p_k)^\top p_k \geq c_2 \nabla f(\boldsymbol{z}_k)^\top p_k, \tag{83}$$

*for constants $0 < c_1 < c_2 < 1$. Specially, $p_k = \Delta \boldsymbol{z}_k$ when line search method satisfies Wolfe condition.*

*Consequently, the sequence of gradients $g_k = \nabla f(\boldsymbol{z}_k)$ converges to zero, that is,*

$$\lim_{k \to \infty} \|g_k\| = 0. \tag{84}$$

**Proof** *By the properties of the equation (1), the first three conditions are already met. Algorithm 2 choose Sinkhorn iteration to decrease $f(\boldsymbol{z})$ if equation (83) is not met by line search. But in Lemma 3 we have already proven that Sinkhorn scaling satisfy Wolfe condition, so Algorithm 2 possess these four conditions.*

*Given that $f(\boldsymbol{z})$ is bounded below, continuously differentiable with a Lipschitz continuous gradient, and the iterates satisfy the Wolfe conditions, we can apply Zoutendijk's Theorem, which states that:*

$$\sum_{k=0}^{\infty} \cos^2 \theta_k \|g_k\|^2 < \infty, \tag{85}$$

*where $\theta_k$ is the angle between the search direction $\Delta \boldsymbol{z}_k$ and the negative gradient $-g_k$, defined by $\cos \theta_k = \frac{-g_k^\top \Delta \boldsymbol{z}_k}{\|g_k\| \|\Delta \boldsymbol{z}_k\|}$. The convergence $\lim_{k \to \infty} \|g_k\| = 0$ follows if $\cos \theta_k$ is uniformly bounded by positive constant.*

*When line search method find the step size $\alpha_k$ satisfies Wolfe condition, the search direction is given by $\Delta \boldsymbol{z}_k = -\hat{H}_k^{-1} g_k$, where $\hat{H}_k$ is a positive definite matrix. From Gershgorin's Circle Theorem we know that the largest eigenvalue of $\hat{H}_k$ is bounded by $2 \max_{i,j}(\boldsymbol{a}_i, \boldsymbol{b}_j)$, so combine with equation (63) we can get that exists constant $c_{line\ search} > 0$ such that $\kappa(\hat{H}_k) \leq c_{line\ search}$.*

*Then we establish a positive lower bound for $\cos \theta_k$:*

$$\cos \theta_k = \frac{-g_k^\top (-\hat{H}_k^{-1} g_k)}{\|g_k\| \|\hat{H}_k^{-1} g_k\|} = \frac{g_k^\top \hat{H}_k^{-1} g_k}{\|g_k\| \|\hat{H}_k^{-1} g_k\|}. \tag{86}$$

*Using the Rayleigh quotient and properties of matrix norms, we bound the numerator and denominator:*

- *Numerator: $g_k^\top \hat{H}_k^{-1} g_k \geq \lambda_{\min}(\hat{H}_k^{-1}) \|g_k\|^2 = \frac{1}{\lambda_{\max}(\hat{H}_k)} \|g_k\|^2$.*

- *Denominator: $\|\hat{H}_k^{-1} g_k\| \leq \|\hat{H}_k^{-1}\|_2 \|g_k\| = \lambda_{\max}(\hat{H}_k^{-1}) \|g_k\| = \frac{1}{\lambda_{\min}(\hat{H}_k)} \|g_k\|$.*

*Combining these inequalities yields:*

$$\cos \theta_k \geq \frac{\frac{1}{\lambda_{\max}(\hat{H}_k)} \|g_k\|^2}{\|g_k\| \cdot \frac{1}{\lambda_{\min}(\hat{H}_k)} \|g_k\|} = \frac{\lambda_{\min}(\hat{H}_k)}{\lambda_{\max}(\hat{H}_k)} = \frac{1}{\kappa(\hat{H}_k)}. \tag{87}$$

*When the line search fails, Algorithm 2 employs a Sinkhorn step as a fallback. Now we demonstrate that this step can also be framed as a preconditioned gradient descent, where the corresponding preconditioner has a uniformly bounded condition number.*

*In the Sinkhorn iteration, the descent direction $\boldsymbol{p}_k$ can be defined by:*

$$\boldsymbol{p}_k = -\tilde{H}_k^{-1} g_k, \tag{88}$$

*where $\tilde{H}_k$ is the matrix given by:*

$$\tilde{H}_k = \beta \begin{pmatrix} \mathrm{diag}(\mathcal{D}(P_k \mathbf{1}_m, \boldsymbol{a})) & \mathbf{0} \\ \mathbf{0} & \mathrm{diag}(\mathcal{D}(P_k^\top \mathbf{1}_n, \boldsymbol{b})) \end{pmatrix}, \tag{89}$$

$\mathcal{D}(\boldsymbol{x}, \boldsymbol{y}) = (\boldsymbol{x} - \boldsymbol{y}) \oslash (\log \boldsymbol{x} - \log \boldsymbol{y})$ and $P_k$ is the transport plan at the $k$-th iteration.

*Let us analyze the properties of $\tilde{H}_k$. Since the target marginals $\boldsymbol{a}$ and $\boldsymbol{b}$ consist of strictly positive elements, the row sums and column sums of the transport plan $P_k$ also remain strictly positive throughout the iterations. Consequently, $\tilde{H}_k$ is a diagonal matrix with strictly positive entries, which ensures it is positive definite.*

*The eigenvalues of $\tilde{H}_k$ are its diagonal entries, specifically $\{\beta \mathcal{D}(P_k \mathbf{1}_m, \boldsymbol{a}))_i\}_i \cup \{\beta \mathcal{D}(P_k^\top \mathbf{1}_n, \boldsymbol{b})_j\}_j$. Therefore, there exist constants $\lambda_{\min}^* > 0$ and $\lambda_{\max}^* < \infty$, independent of $k$, such that all eigenvalues of $\tilde{H}_k$ lie within the interval $[\lambda_{\min}^*, \lambda_{\max}^*]$. This implies that its condition number is uniformly bounded:*

$$\kappa(\tilde{H}_k) = \frac{\lambda_{\max}(\tilde{H}_k)}{\lambda_{\min}(\tilde{H}_k)} \leq \frac{\lambda_{\max}^*}{\lambda_{\min}^*} =: c_{Sinkhorn}. \tag{90}$$

*Substituting this result into the same inequality for $\cos \theta_k$ as in the former case, we obtain:*

$$\cos \theta_k \geq \frac{1}{\kappa(\tilde{H}_k)} \geq \frac{1}{c_{Sinkhorn}} > 0. \tag{91}$$

*Since $\kappa(\hat{H}_k) \leq c_{line\ search}$, $\kappa(\tilde{H}_k) \leq c_{Sinkhorn}$, let $c = \max(c_{line\ search}, c_{Sinkhorn})$, we have $\cos \theta_k \geq \frac{1}{c} > 0$, which provides a uniform positive lower bound. Applying this to Zoutendijk's condition:*

$$\infty > \sum_{k=0}^{\infty} \cos^2 \theta_k \|g_k\|^2 \geq \sum_{k=0}^{\infty} \frac{1}{c^2} \|g_k\|^2 = \frac{1}{c^2} \sum_{k=0}^{\infty} \|g_k\|^2. \tag{92}$$

*This implies that the series $\sum_{k=0}^{\infty} \|g_k\|^2$ must converge. Therefore,*

$$\lim_{k \to \infty} \|g_k\|^2 = 0, \tag{93}$$

*which implies:*

$$\lim_{k \to \infty} \|g_k\| = 0. \tag{94}$$

**Theorem 3 (Global Convergence)** *The sequence $\{\boldsymbol{z}_k\}$ generated by Algorithm 2 converges globally to the minimizer $\boldsymbol{z}^*$.*

**Proof** *Now we begin to prove global convergence. Let $\boldsymbol{z}^*$ be the global optimum of (14). Since $f$ and $\nabla f$ are both Lip-continuous, $\exists\, c_1, c_2 > 0$ such that*

$$|f(\boldsymbol{z}_k) - f(\boldsymbol{z}^*)| \leq c_1 \|\boldsymbol{z}_k - \boldsymbol{z}^*\|, \tag{95}$$

$$\|\nabla f(\boldsymbol{z}_k) - \nabla f(\boldsymbol{z}^*)\| \leq c_2 \|\boldsymbol{z}_k - \boldsymbol{z}^*\|. \tag{96}$$

*Firstly, Tang and Qiu (2024) show that the smallest eigenvalue of the Hessian matrix of the function $f$ define by (14) has a positive lower bound, i.e. $\exists\, c_3 > 0$, $\lambda_{\min}(H(\boldsymbol{z})) \geq c_3$, so we can get that*

$$(\nabla f(\boldsymbol{z}_k) - \nabla f(\boldsymbol{z}^*))^\top (\boldsymbol{z}_k - \boldsymbol{z}^*) \geq c_3 \|\boldsymbol{z}_k - \boldsymbol{z}^*\|^2, \tag{97}$$

*which implies*

$$\|\nabla f(\boldsymbol{z}_k)\| = \|\nabla f(\boldsymbol{z}_k) - \nabla f(\boldsymbol{z}^*)\| \geq c_3 \|\boldsymbol{z}_k - \boldsymbol{z}^*\|. \tag{98}$$

*On the other hand, by Taylor's theorem,*

$$f(\boldsymbol{z}_k) = f(\boldsymbol{z}^*) + (\nabla f(\boldsymbol{z}_k))^\top (\boldsymbol{z}_k - \boldsymbol{z}^*) + \frac{1}{2}(\boldsymbol{z}_k - \boldsymbol{z}^*)^\top H(\tilde{\boldsymbol{z}})(\boldsymbol{z}_k - \boldsymbol{z}^*), \tag{99}$$

*where $\tilde{z}$ is a point between $\boldsymbol{z}_k$ and $\boldsymbol{z}^*$, so we have*

$$|f(\boldsymbol{z}_k) - f(\boldsymbol{z}^*)| \geq \frac{c_3}{2} \|\boldsymbol{z}_k - \boldsymbol{z}^*\|^2. \tag{100}$$

*Since $f$ is non-increase and has lower bound, $\exists\, \epsilon \geq 0$ such that*

$$\lim_{k \to \infty} |f(\boldsymbol{z}_k) - f(\boldsymbol{z}^*)| = \epsilon. \tag{101}$$

*Combine (95) and (101), for sufficient large $k$ we have*

$$\frac{1}{2} c_1^{-1} \epsilon \leq c_1^{-1} |f(\boldsymbol{z}_k) - f(\boldsymbol{z}^*)| \leq \|\boldsymbol{z}_k - \boldsymbol{z}^*\| \leq c_3^{-1} \|\nabla f(\boldsymbol{z}_k)\|. \tag{102}$$

*Combine (96), (100) and (101), for sufficient large $k$ we have*

$$\frac{c_3}{2c_2^2} \|\nabla f(\boldsymbol{z}_k)\|^2 \leq \frac{c_3}{2} \|\boldsymbol{z}_k - \boldsymbol{z}^*\|^2 \leq |f(\boldsymbol{z}_k) - f(\boldsymbol{z}^*)| \to \epsilon. \tag{103}$$

*Since our line search method and Sinkhorn iteration both satisfy Wolfe condition, applying Lemma 4 we get that $\epsilon = 0$. Combine (102) and (103) with squeeze theorem, we have*

$$\lim_{k \to \infty} \|\boldsymbol{z}_k - \boldsymbol{z}^*\| = 0. \tag{104}$$

**Theorem 4** *Algorithm 3 converges globally to the solution of the corresponding entropic OT.*

**Proof** *The global convergence of the PSN algorithm follows from the global convergence of its two components: the Proximal-Sinkhorn (PS) and Proximal-Newton (PN). The convergence of PN is established by Theorem 3, while that of PS is guaranteed by the global convergence of the Sinkhorn algorithm Cuturi (2013). Hence, PSN is globally convergent.*

## E  THE SPECTRAL DISTRIBUTION OF THE HESSIAN MATRIX

The following Lemma and Theorem characterize the convergence behavior of CG. Lemma 5 shows that the error at each iteration of CG is bounded by a polynomial $q_k(\lambda)$, and Theorem 5 reveals the connection between the polynomial $q_k(\lambda)$ and the spectral distribution of the Hessian matrix $H$ while solving $H\Delta\boldsymbol{z} = -\nabla f$. Furthermore, Theorem 5 also demonstrates that the error vector in each iteration of the CG method remains bounded by the polynomial $q_k(\lambda)$, even after excluding a small number of eigenvalues at one or both extremes. This reveals that the CG method's convergence rate is closely tied to the eigenvalue distribution of the matrix $H$, potentially leading to superlinear rates.

**Lemma 5 ((Saad, 2003))** *The approximate solution $\boldsymbol{x}_k$ generated by the CG method of solving $H\boldsymbol{x} = \boldsymbol{b}$ at step $k$ satisfies*

$$\|\boldsymbol{x}_k - \boldsymbol{x}^*\|_H \leqslant \min_{q_k \in \mathcal{P}_k^{(0)}} \max_{\lambda \in \lambda(H)} |q_k(\lambda)| \|\boldsymbol{x}_0 - \boldsymbol{x}^*\|_H, \tag{105}$$

*where $\boldsymbol{x}^*$ denotes the exact solution of the linear system, $\|\cdot\|_H$ represents the $H$-norm, $\mathcal{P}_k^{(0)}$ represents the set of all real coefficient polynomials of degree not exceeding $k$ that satisfy $q_k(0) = 1$.*

**Theorem 5 ((Hestenes et al., 1952))** *Considering the error vectors $\boldsymbol{u}_k = \boldsymbol{x}_k - \boldsymbol{x}^*$ defined by Lemma 5, we establish the following bounds:*

*(1) for $\lambda(H) \subset \{\lambda_1, \cdots, \lambda_p\} \cup [a_1, b_1]$, where $\lambda_i \in \lambda(H)$, $\lambda_i < a_1$, we have*

$$\|\boldsymbol{u}_k\|_H \leqslant F(a_1, b_1; k-p) \prod_{i=1}^{p} \frac{b_1}{\lambda_i} \|\boldsymbol{u}_0\|_H; \tag{106}$$

*(2) for $\lambda(H) \subset [a_2, b_2] \cup \{\lambda_1', \cdots, \lambda_p'\}$, where $\lambda_i' \in \lambda(H)$, $\lambda_i' > b_2$, we have*

$$\|\boldsymbol{u}_k\|_H \leqslant F(a_2, b_2; k-p) \|\boldsymbol{u}_0\|_H; \tag{107}$$

*(3) for $\lambda(H) \subset \{\lambda_1, \cdots, \lambda_p\} \cup [a_3, b_3] \cup \{\lambda_1', \cdots, \lambda_p'\}$, where $\lambda_i, \lambda_i' \in \lambda(H)$, $\lambda_i' > b_3$, $\lambda_i < a_3$, we have*

$$\|\boldsymbol{u}_k\|_H \leqslant \frac{1}{4} \prod_{i=1}^{p} \frac{\lambda_i'}{\lambda_i} \left(1 - \frac{\lambda_i}{\lambda_i'}\right)^2 F(a_3, b_3; k-2) \|\boldsymbol{u}_0\|_H, \tag{108}$$

*where $F(a, b; k) = \left(T_k\left(\frac{b+a}{b-a}\right)\right)^{-1}$, and $T_k(x)$ denotes Chebyshev polynomial of degree $k$.*

## F  PERFORMANCE ON DIFFERENT ENTROPIC PARAMETERS

We evaluate PSN, SNS, SSNS, and the Newton method across entropic parameters $\beta \in \{100, 500, 3000\}$ covering both low and high regularization. Figure 6 indicates that our proposed algorithm, PSN, exhibits superior robustness and effectiveness when $\beta$ is large. For smaller values of $\beta$, PSN reduces to a combination of the proximal point method and Sinkhorn scaling, and outperforms other Newton-based algorithms as well as the log-Sinkhorn method.

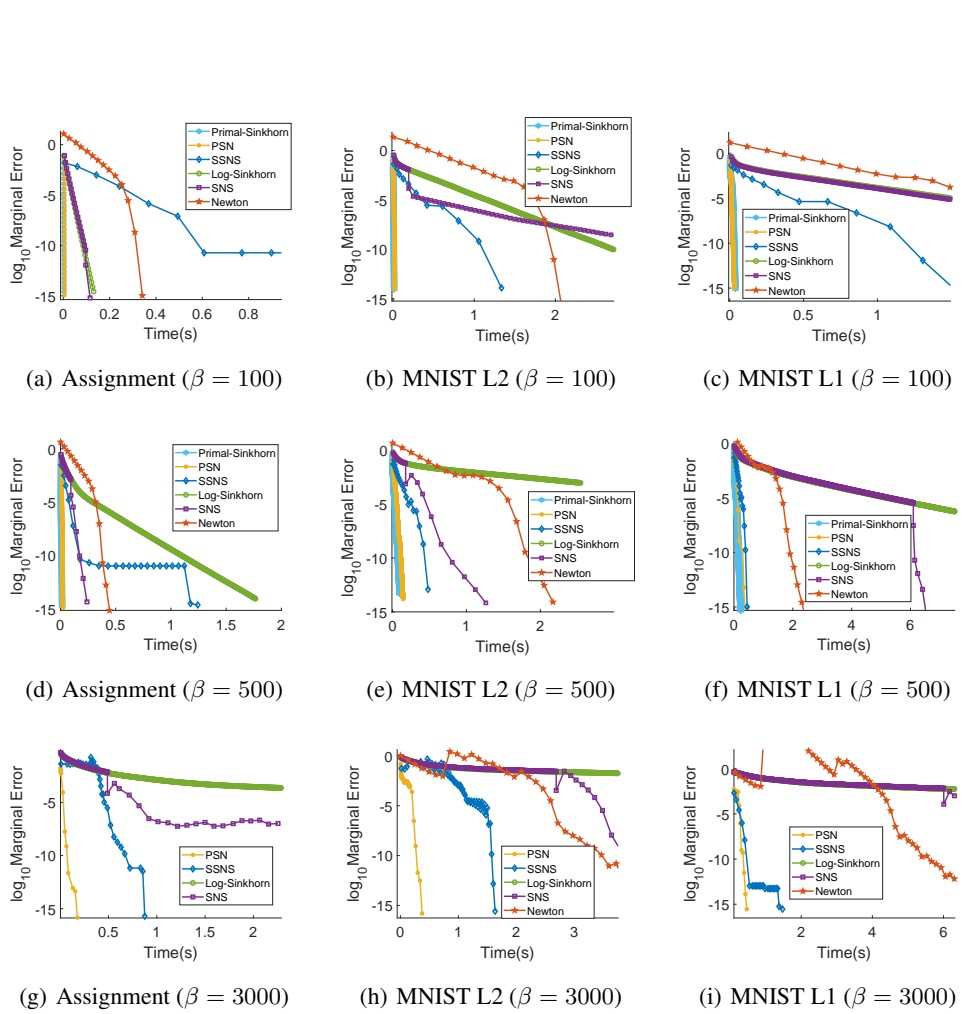

Figure 6: Performance on different $\beta$.

## G  COMPARISON WITH OTHER CLASSIC FIRST ORDER ALGORITHMS

In this section, we compare our algorithm against several well-established first order optimization methods, including Limited-memory Broyden–Fletcher–Goldfarb–Shanno algorithm (L-BFGS, (Liu and Nocedal, 1989)) for the quasi-Newton method, the adaptive primal-dual accelerated gradient descent method (ADPAGD, (Dvurechensky et al., 2018)) as an accelerated first-order approach, and the Sinkhorn method implemented as a block coordinate descent method (BCD, (Luo and Tseng, 1992)). We adopted the C++ implementation by Tang and Qiu (2024) for the baseline algorithms[4].

Evaluated across our three datasets, our algorithm, as depicted in Figure 7, consistently delivers superior performance, significantly outperforming these classic optimization techniques in all situations.

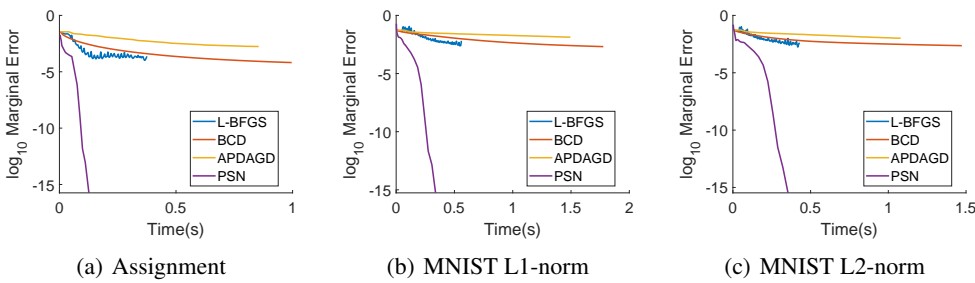

|  (a) Assignment  |  (b) MNIST L1-norm  |  (c) MNIST L2-norm  |

Figure 7: Other classic algorithms.

## H  TRUNCATION DETAILS

We explore the sparsity of the Hessian matrix under various truncation thresholds and visualize the results in Figure 8.

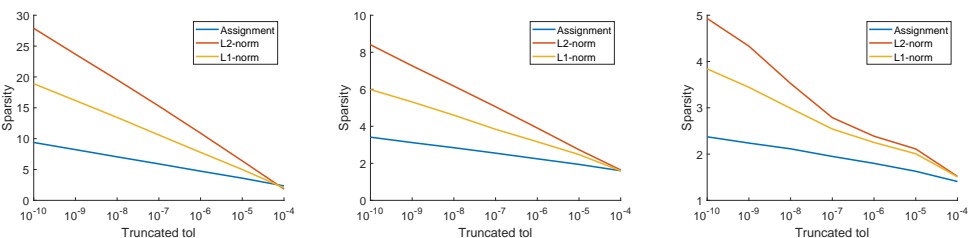

Figure 8: Relationship between truncation threshold and Sparsity. The three subfigures correspond to $\beta = \{1000, 2000, 4000\}$, respectively. The $x$-axis denotes the truncation threshold, while the $y$-axis represents the Hessian matrix sparsity as a multiple of $1/n$.

Figure 8 demonstrates that the truncated matrix exhibits significant sparsity. Moreover, under the same truncation threshold, larger values of $\beta$ lead to a sparser matrix.

Interestingly, the relationship between the truncation threshold and the resulting sparsity level is nearly linear. Consequently, when the sparsified matrix contains fewer than $6n$ nonzero entries, we can readily determine the truncation threshold that yields approximately $6n$ nonzeros.

---

[4]https://github.com/yixuan/regot-python.

## I  DETAILED RESULTS OF FIGURE 5

| Case | Error | IP-EOT | | Primal-Sinkhorn | | Log-Sinkhorn | | PSN | | SSNS | | SNS | | Newton | | SSNS/PSN |
|---|---|---|---|---|---|---|---|---|---|---|---|---|---|---|---|---|
| | | # iters | time | # iters | time | # iters | time | # iters | time | # iters | time | # iters | time | # iters | time | |
| Assignment $n=500$ | $10^{-4}$ | 16 | 0.01s | 12 | 0.001s | 12 | 0.06s | 11 | 0.005s | 6 | 0.14s | 12 | 0.06s | 12 | 0.32s | 28× |
| | $10^{-6}$ | 148 | 0.08s | 25 | 0.002s | 25 | 0.10s | 23 | 0.006s | 7 | 0.20s | 201 | 0.09s | 13 | 0.35s | 33× |
| | $10^{-8}$ | 1466 | 0.76s | 43 | 0.003s | 43 | 0.19s | 42 | 0.007s | 8 | 0.30s | 20+2 | 0.11s | 14 | 0.38s | 42× |
| MNIST L2 $n=784$ | $10^{-4}$ | 39 | 0.04s | 176 | 0.02s | 176 | 1.48s | 109 | 0.02s | 10 | 0.22s | 20+1 | 0.17s | 13 | 1.45s | 11× |
| | $10^{-6}$ | 111 | 0.12s | 392 | 0.04s | 392 | 3.33s | 324 | 0.03s | 13 | 0.38s | 20+4 | 0.4s | 14 | 1.57s | 13× |
| | $10^{-8}$ | 671 | 0.77s | 608 | 0.05s | 608 | 5.16s | 540 | 0.04s | 15 | 0.54s | 20+9 | 0.82s | 15 | 1.69s | 14× |
| MNIST L1 $n=784$ | $10^{-4}$ | 57 | 0.06s | 148 | 0.02s | 148 | 1.25s | 114 | 0.04s | 10 | 0.16s | 148 | 1.26s | 13 | 1.51s | 4× |
| | $10^{-6}$ | 296 | 0.32s | 409 | 0.05s | 409 | 3.50s | 354 | 0.07s | 15 | 0.36s | 409 | 3.45s | 15 | 1.75s | 5.1× |
| | $10^{-8}$ | 2948 | 3.42s | 674 | 0.07s | 674 | 5.75s | 617 | 0.13s | 16 | 0.42s | 674 | 5.69s | 16 | 1.87s | 3.2× |

Table 2: Comparison of iterations and runtime (in seconds) at different accuracy levels with $\beta = 300$. Underlined numbers indicate the iteration count of Newton methods, while non-underlined numbers correspond to the iteration count of IP-EOT/Sinkhorn.

| Case | Error | IP-EOT | | Log-Sinkhorn | | PSN | | SNS | | SSNS | | Newton | | SSNS/PSN |
|---|---|---|---|---|---|---|---|---|---|---|---|---|---|---|
| | | # iters | time | # iters | time | # iters | time | # iters | time | # iters | time | # iters | time | |
| Assignment $n=500$ | $10^{-4}$ | 26 | 0.021s | 73 | 0.26s | 24+1 | 0.03s | 20+1 | 0.09s | 11 | 0.2s | | | 6.7× |
| | $10^{-6}$ | 236 | 0.15s | 654 | 2.23s | 24+2 | 0.04s | 20+5 | 0.28s | 13 | 0.26s | break | | 6.5× |
| | $10^{-8}$ | 2335 | 2.93s | 9679 | 33.6s | 24+3 | 0.06s | 20+6 | 0.38s | 14 | 0.30s | | | 5× |
| MNIST L2 $n=784$ | $10^{-4}$ | 77 | 0.09s | 770 | 4.68s | 52+1 | 0.14s | 20+13 | 1.0s | 19 | 0.42s | 14 | 1.64s | 3.0× |
| | $10^{-6}$ | 357 | 0.41s | 1713 | 10.0s | 52+3 | 0.20s | 20+16 | 1.30s | 26 | 0.78s | 16 | 1.92s | 3.9× |
| | $10^{-8}$ | 2776 | 4.90s | 3128 | 18.0s | 52+5 | 0.26s | 20+17 | 1.41s | 28 | 0.88s | 17 | 2.05s | 7.9× |
| MNIST L1 $n=784$ | $10^{-4}$ | 111 | 0.14s | 589 | 3.39s | 52+1 | 0.13s | 589 | 3.32s | 24 | 0.44s | 15 | 1.86s | 3.4× |
| | $10^{-6}$ | 427 | 0.54s | 1540 | 8.72s | 52+3 | 0.18s | 700+1 | 3.94s | 29 | 0.68s | 17 | 2.11s | 3.8× |
| | $10^{-8}$ | 3152 | 8.94s | 3052 | 17.3s | 52+5 | 0.24s | 700+3 | 4.18s | 31 | 0.78s | 18 | 2.23s | 3.3× |

Table 3: Comparison of iterations and runtime (in seconds) at different accuracy levels and $\beta = 200 \ln n$. Underlined numbers indicate the iteration count of Newton methods, while non-underlined numbers correspond to the iteration count of IP-EOT/Sinkhorn. The last column reports the SSNS/PSN ratio on time, quantifying the performance gain of PSN over the second-best method.

| Case | Error | IP-EOT | | Log-Sinkhorn | | PSN | | SSNS | | SNS | | Newton | | SSNS/PSN | SNS/PSN |
|---|---|---|---|---|---|---|---|---|---|---|---|---|---|---|---|
| | | # iters | time | # iters | time | # iters | time | # iters | time | # iters | time | # iters | time | | |
| Assignment $n=4000$ | $10^{-4}$ | 13 | 0.69s | 8 | 2.10s | 34 | 2.29s | 7 | 40.2s | 8 | 2.32s | 10 | 26.4s | 16× | 1.0× |
| | $10^{-6}$ | 120 | 6.23s | 16 | 4.08s | 34+1 | 3.26s | 9 | 89.8s | 16 | 4.65s | 13 | 36.6s | 27× | 1.4× |
| | $10^{-8}$ | 1189 | 63.1s | 27 | 6.44s | 34+2 | 4.24s | 12 | 204s | 20+1 | 5.33s | 15 | 49.1s | 48× | 1.3× |
| MNIST L2 $n=4096$ | $10^{-4}$ | 85 | 4.61s | 956 | 246s | 68 | 5.15s | 29 | 17.7s | 20+6 | 15.5s | 15 | 238s | 3.4× | 3.0× |
| | $10^{-6}$ | 400 | 21.6s | 2133 | 489s | 68+3 | 7.06s | 42 | 29.4s | 20+9 | 23.2s | 19 | 310s | 4.2× | 3.3× |
| | $10^{-8}$ | 693 | 37.7s | 3311 | 761s | 68+4 | 8.01s | 46 | 33.4s | 20+31 | 77.6s | 20 | 328s | 4.2× | 9.7× |
| MNIST L1 $n=4096$ | $10^{-4}$ | 124 | 6.94s | 757 | 189s | 68+2 | 5.71s | 32 | 20.2s | 700 | 164s | 16 | 259s | 3.5× | 29× |
| | $10^{-6}$ | 484 | 26.8s | 1974 | 444s | 68+4 | 7.53s | 42 | 28.0s | 700+1 | 165s | 20 | 328s | 3.7× | 22× |
| | $10^{-8}$ | 3189 | 207s | 3191 | 718s | 68+5 | 8.37s | 48 | 33.4s | 700+4 | 178s | 21 | 345s | 4.0× | 21× |

Table 4: Comparison of iterations and runtime (in seconds) at different accuracy levels and $\beta = 200 \ln n$. Underlined numbers indicate the iteration count of Newton methods, while non-underlined numbers correspond to the iteration count of IP-EOT/Sinkhorn.

## J    DISCUSSION OF THE SWITCHING PARAMETER

The choice of $\lambda$ aims to select the more efficient option between Proximal-Sinkhorn (PS) and Proximal-Newton (PN). Accordingly, this section presents the efficiency of PS and PN under different values of $\beta$, together with the corresponding changes in the sparsity of $\hat{H}$, providing guidance for choosing an appropriate $\lambda$.

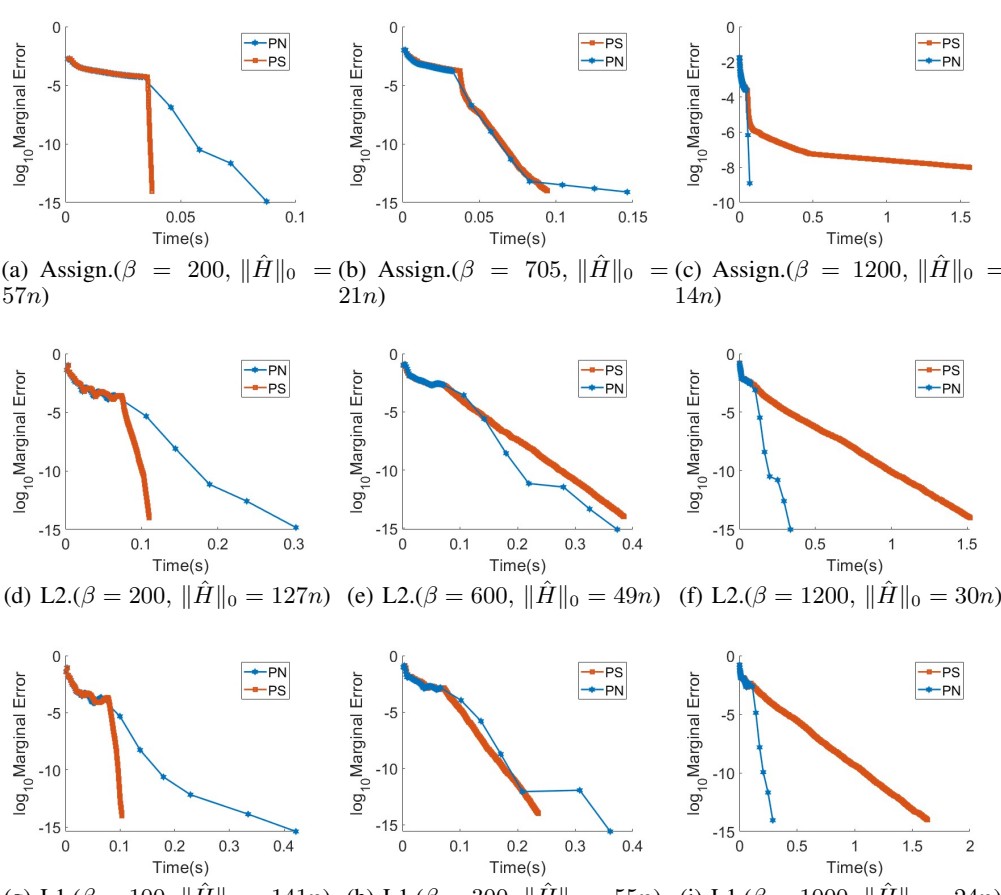

(a) Assign.($\beta = 200$, $\|\hat{H}\|_0 = 57n$) (b) Assign.($\beta = 705$, $\|\hat{H}\|_0 = 21n$) (c) Assign.($\beta = 1200$, $\|\hat{H}\|_0 = 14n$)

(d) L2.($\beta = 200$, $\|\hat{H}\|_0 = 127n$) (e) L2.($\beta = 600$, $\|\hat{H}\|_0 = 49n$) (f) L2.($\beta = 1200$, $\|\hat{H}\|_0 = 30n$)

(g) L1.($\beta = 100$, $\|\hat{H}\|_0 = 141n$) (h) L1.($\beta = 300$, $\|\hat{H}\|_0 = 55n$) (i) L1.($\beta = 1000$, $\|\hat{H}\|_0 = 24n$)

Figure 9: Performance of PS and PN algorithm on three datasets with different $\beta$ value.

To discuss the selection of $\lambda$, we evaluate PS and PN under different values of $\beta$ to explore the relation between the number of non-zero elements $|\hat{H}|_0$ and performance, we specifically selected three sets of $\beta$ such that the three scenarios—namely, PS being superior, the two being comparable, and PN being superior—are clearly demonstrated. As shown in Fig. 9, as $\beta$ increases, the number of non-zero elements $|\hat{H}|_0$ gradually decreases, and PN progressively outperforms PS. Interestingly, under both L1 and L2 settings, when PS and PN achieve comparable time performance, their sparsity levels are similar even though the values of $\beta$ differ, which validates the effectiveness of using $\lambda$ as the switching parameter.

## K    COMPARISON WITH EXISTING FIRST ORDER METHODS

To illustrate the efficiency of our Proximal-Sinkhorn method (PS), we compare it with two classical entropic solvers: log-Sinkhorn and Epsilon-Scaling method[5]. We compare the algorithms for $\beta$ in the

---

[5]Schmitzer, B. (2016). Stabilized Sparse Scaling Algorithms for Entropy Regularized Transport Problems. arXiv:1610.06519.

range (100, 700), since for larger values second-order methods gain a clear advantage (as shown in Figure 9).

As shown in Figure 10, the error curve of the epsilon-scaling method exhibits an initial plateau before decreasing, because it requires a warm-up phase. Our PS method does not suffer from this issue and consistently achieves superior performance.

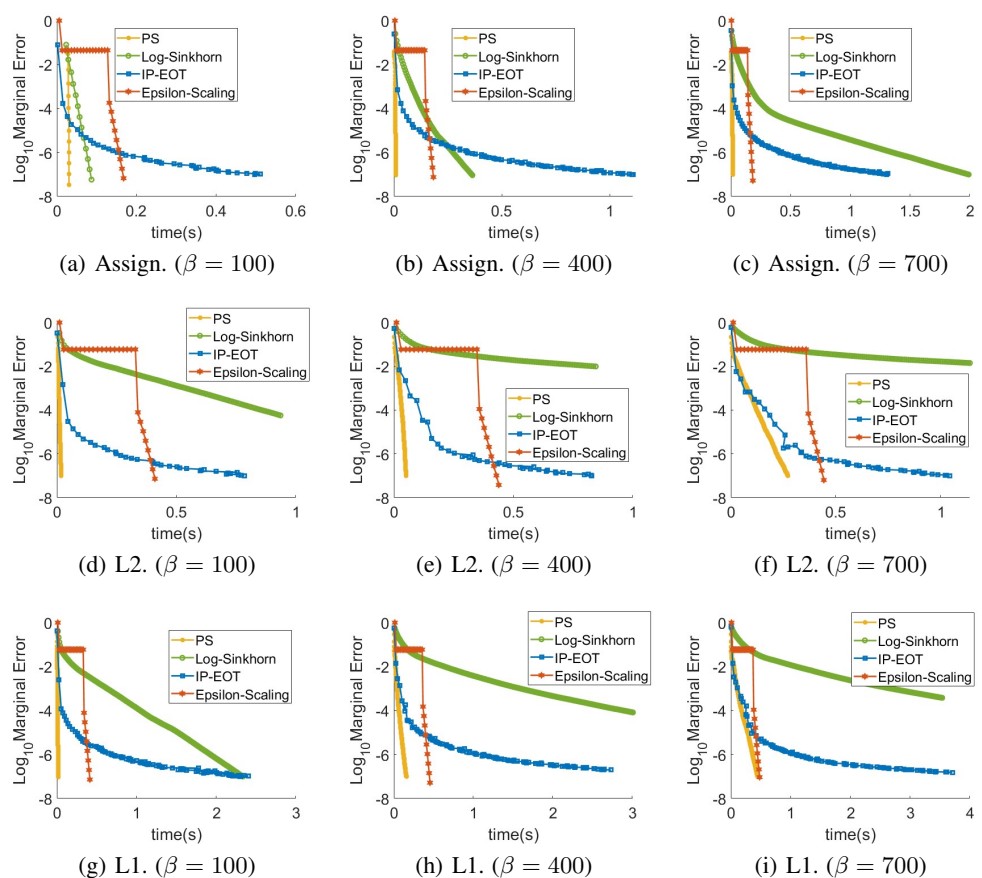

Figure 10: Comparison of four first-order algorithms.

## L    EVALUATION ON IMAGENET

We consider an OT problem between two categories of images from the ImageNet dataset. We use a subset of ImageNet from the Imagenette GitHub repository[6], which contains ten image classes. Approximately 1000 images per class are selected. Each image is mapped to a 30-dimensional feature vector by first passing it through a ResNet18 network to obtain a 512-dimensional embedding, followed by dimensionality reduction via principal component analysis.

Let $x_i \in \mathbb{R}^{30}$ denote the feature vector of an image in the first category for $i = 1, \ldots, n$, and let $y_j \in \mathbb{R}^{30}$ denote the feature vector of an image in the second category for $j = 1, \ldots, m$. We set $\boldsymbol{a} = n^{-1}\mathbf{1}_n$, $\boldsymbol{b} = m^{-1}\mathbf{1}_m$, and define the cost matrix either as $C_{ij} = \|x_i - y_j\|_1$ or $C_{ij} = \|x_i - y_j\|^2$. The problem size is $n \approx 1000$.

As shown in Figure 11, both Newton and SNS encounter failures in at least five cases, and even SSNS fails in one instance. In contrast, PSN achieves a perfect success rate across all test cases, demonstrating reliable performance and high efficiency in every setting.

---

[6]https://github.com/fastai/imagenette

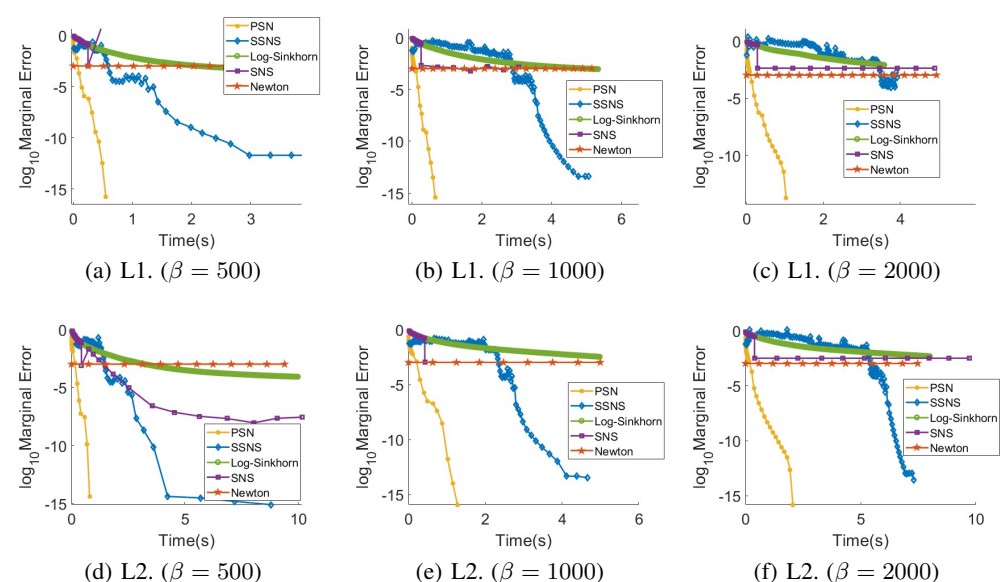

Figure 11: Performance on the ImageNet dataset.

## M  DISCUSSION OF THE PROXIMAL STEP SIZE

This section examines the sensitivity of PSN to the proximal step size $\Delta\beta$. According to Theorem 1, the proximal step size $\Delta\beta$ influences only the initialization of the second stage. When the second stage is achieved by Sinkhorn scaling, $\Delta\beta$ does not affect the convergence rate. Therefore, this section focuses on the sensitivity of the Proximal–Newton (PN) method to the choice of $\Delta\beta$.

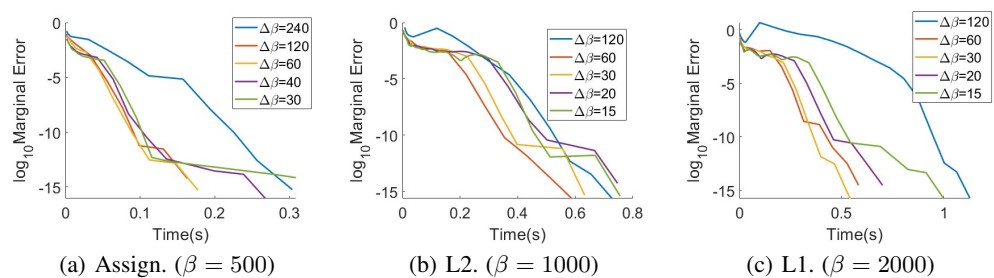

Figure 12: Impact of the proximal step size $\Delta\beta$ on proximal-newton method.

As illustrated in Fig. 12, the PN algorithm exhibits considerable robustness to the choice of proximal step size, consistently achieving stable convergence across different datasets and values of $\beta$. The convergence behavior remains highly similar across a wide range of step sizes, except when $\Delta\beta$ is excessively large—that is, when the number of IP-EOT iterations $\ell$ is particularly small. Even in such cases, the algorithm only suffers from a slightly slower initial convergence rate, with the total computation time remaining comparable. On the other hand, when $\beta$ is large and the proximal step size is small (corresponding to a large number of IP-EOT iterations $\ell$), the computational cost of IP-EOT increases somewhat. Overall, as long as the proximal step size is chosen to ensure the numerical stability of the IP-EOT phase, the algorithm PSN achieves favorable convergence performance.

# N VARIABILITY OF THE METHODS UNDER DIFFERENT RANDOM REALIZATIONS

For each scenario presented in the Figure 5 (a), (d) and (g), we conducted twenty independent random trials and plotted the results with variance. Figure 13 demonstrates that Newton and PSN exhibit small fluctuations under randomness, whereas SNS and SSNS show much larger variability.

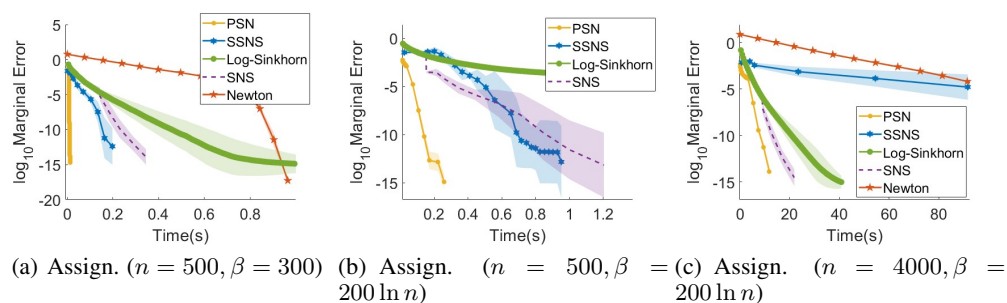

(a) Assign. ($n = 500, \beta = 300$)  (b) Assign. ($n = 500, \beta = 200 \ln n$)  (c) Assign. ($n = 4000, \beta = 200 \ln n$)

Figure 13: Performance under random realizations on the linear assignment tasks.

# O ZOOM-IN OF SOME FIGURES

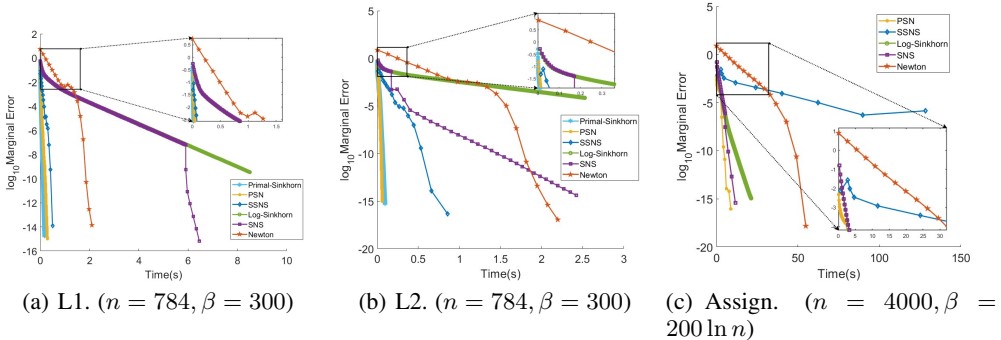

(a) L1. ($n = 784, \beta = 300$)  (b) L2. ($n = 784, \beta = 300$)  (c) Assign. ($n = 4000, \beta = 200 \ln n$)

Figure 14: Zoom-in plots.

# P TIME PER ITERATION

| Case | IP-EOT | Log-Sinkhorn | PS | Epsilon-Scaling |
|---|---|---|---|---|
| Assignment $n = 500$ | 5e-4s | 5e-3s | 6e-4s | 4e-3s |
| MNIST L2 $n = 784$ | 1e-3s | 8e-3s | 6e-4s | 0.014s |
| MNIST L1 $n = 784$ | 1e-3s | 9e-3s | 6e-4s | 0.014s |

Table 5: Time per iteration of four first-order methods under $\beta = 300$.

| Case | PN (Newton part) | SSNS | SNS (Newton part) | Newton |
|---|---|---|---|---|
| Assignment $n = 500$ | 0.03s | 0.04s | 0.1s | break |
| MNIST L2 $n = 784$ | 0.04s | 0.03s | 0.2s | 0.1s |
| MNIST L1 $n = 784$ | 0.03s | 0.03s | 0.3s | 0.15s |

Table 6: Time per iteration of four second-order methods under $\beta = 1200$.

## Q    SPECTRAL ESTIMATION

By invoking Gershgorin's Circle Theorem, the eigenvalues of the Hessian matrix $H$, defined in equation (15), satisfy

$$\lambda(H) \in \left( \bigcup_{i=1}^{n} [0, 2\beta\boldsymbol{a}_i] \right) \bigcup \left( \bigcup_{j=1}^{n} [0, 2\beta\boldsymbol{b}_j] \right) = [0, 2\beta M_{max}], \tag{109}$$

where $M = \max_{i,j}(\boldsymbol{a}_i, \boldsymbol{b}_j)$. Let $\Delta a_i = a_i - \sum_j \widehat{P}_{ij}$ be the source mass loss, analogously for $\Delta b_j = b_j - \sum_i \widehat{P}_{ij}$ as target mass loss. Let $\Delta_{\min} = \min_{i,j}(\Delta a_i, \Delta b_j)$, the eigenvalues $\lambda$ of $\hat{H}$ satisfy the following bounds

$$\beta\Delta_{\min} \le \lambda(\hat{H}) \le 2\beta M - \beta\Delta_{\min}. \tag{110}$$

To illustrate the behavior under different problem scales, we use the linear assignment problem to validate the range.

**Upper bound.** In the context of the linear assignment problem, we set $\boldsymbol{a} = \boldsymbol{b} = \boldsymbol{1}/n$, which yields an upper bound of $2\beta/n$ for the largest eigenvalue. As illustrated in Figure 15, the largest eigenvalue of $\hat{H}$ consistently adheres to this theoretical bound across all tested scenarios.

**Lower bound.** As illustrated in Figure 16, the smallest eigenvalue reaches a steady state within a negligible number of initial iterations and remains virtually constant thereafter, regardless of the problem size $n$ or the regularization parameter $\beta$. It demonstrates that our method effectively bounds the smallest eigenvalue away from zero, thereby ensuring numerical stability across a broad parameter space.

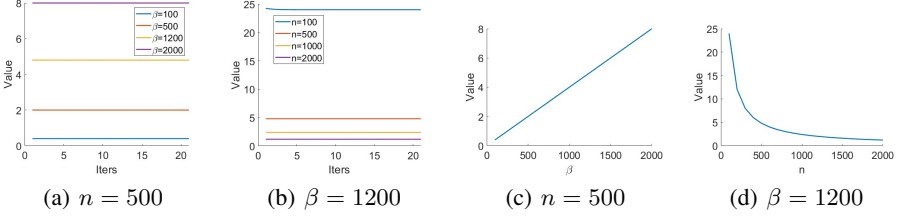

    (a) $n = 500$        (b) $\beta = 1200$        (c) $n = 500$        (d) $\beta = 1200$

Figure 15: The largest eigenvalues of $\hat{H}$ with different $n$ and $\beta$.

As for the feature dimension $d$, Figure 17 shows that the upper bound of the eigenvalues is almost unaffected by $d$, while the lower bound experiences slight perturbations.

Moreover, both the largest and smallest eigenvalues of $\hat{H}$ are almost independent of the iteration, which indirectly confirms the robustness of our sparse Newton method.

## R    LARGER SCALE PROBLEM

We compared our algorithm with other algorithms on the linear assignment dataset with $n = 10^4$ and MNIST L2-norm problem with $n = 10^5$. Figure 18 (a) demonstrates the superiority of our PSN.

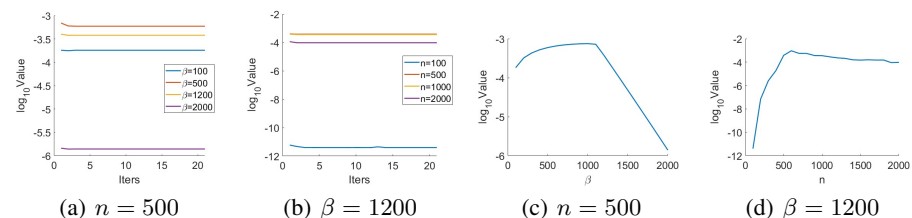

Figure 16: The smallest eigenvalues of $\hat{H}$ with different $n$ and $\beta$.

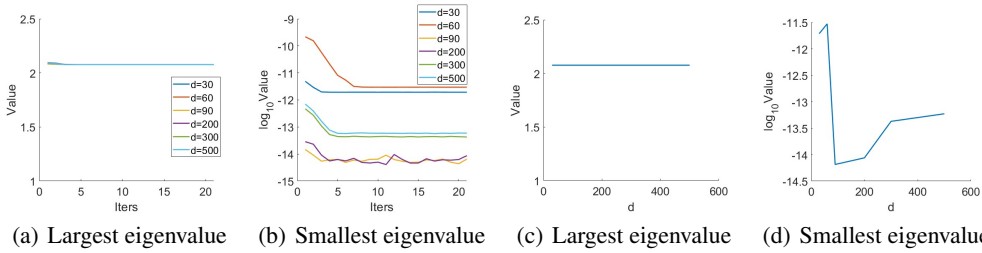

Figure 17: The largest and smallest eigenvalue of $\hat{H}$ with different feature dimension $d$ under $n = 963$ and $\beta = 1000$.

To evaluate the scalability of the algorithm, we conducted experiments on a problem of size $n = 10^5$. To address memory limitations, we implemented a scalable version of PSN with an $O(n)$ memory footprint. In this version, we avoid explicitly storing any matrices with $O(n^2)$ nonzeros, such as $C$, $\Delta\beta$, and $\hat{P}$ in Algorithm 1 ($\hat{H}$ with only $O(n)$ nonzeros can be stored directly). When necessary, these matrices are computed on the fly from relevant vectors and accessed in blocks. Although this approach introduces considerable additional computational overhead, it enables the algorithm to be applied to larger-scale problems and demonstrates its robustness. The results are shown in Table 7 and Figure 18 (b). The parameters are set as follows: $\beta = 10^4$ and $\ell = 100$.

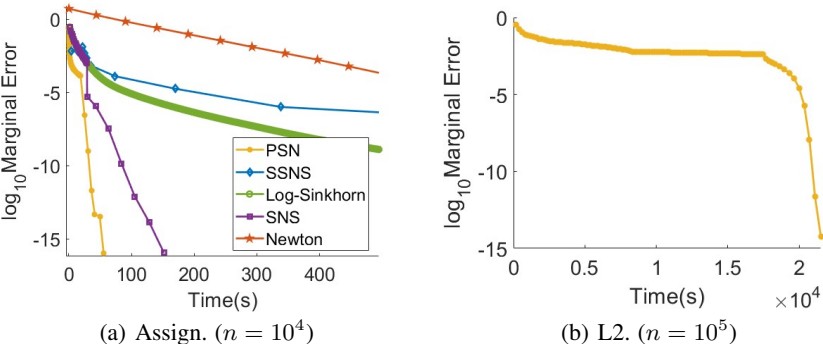

Figure 18: Larger scale problems.

| Marginal error | $10^{-2}$ | $10^{-4}$ | $10^{-6}$ | $10^{-8}$ | $10^{-12}$ | $10^{-14}$ |
|---|---|---|---|---|---|---|
| Time | 799s | 18321s | 20361s | 20727s | 21115s | 21526s |

Table 7: Runtime (in seconds) on a problem of size $n = 10^5$.

## THE USE OF LARGE LANGUAGE MODEL (LLM)

We used a large language model (LLM) solely for grammar checking and stylistic polishing of the manuscript text. The LLM did not generate scientific content, ideas, experiments, analyses, code, or results. All technical contributions, claims, and conclusions are the authors' own. The authors verified all edited text and remain fully responsible for the content. No confidential, proprietary, or personally identifying information was provided to the LLM.

