# A PROXIMAL-SINKHORN-NEWTON METHOD FOR ENTROPIC OPTIMAL TRANSPORT

## ABSTRACT

Entropic optimal transport (OT) enables efficient distribution alignment through the Sinkhorn method. However, it suffers from numerical instability and slow convergence under weak entropic regularization. We propose a two-stage framework that establishes an inexact-to-exact paradigm to address these challenges. The first stage employs an inexact proximal point method to decompose the entropic OT into simpler subproblems, yielding approximate solutions with superior numerical stability. The second stage employs a sparse Newton method with global convergence and a locally quadratic rate to refine the approximate solutions. Compared to previous Newton-based algorithms, it accelerates updates and prevents the objective ~~score~~ value from plateauing during optimization. With numerical instability handled in the first stage, Sinkhorn ~~iterations~~ scaling can provide an alternative to the Newton method under relatively heavy entropic regularization. The ~~yielding~~ resulting Proximal-Sinkhorn-Newton method enjoys the strengths of three approaches and outperforms the baselines across various regularizations and error tolerances~~, achieving over 20× speedups on some problems compared to other Newton-based methods~~.

## 1 INTRODUCTION

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

$ ~~(interpretable as result from insufficient Sinkhorn iterations) by making~~ by explicitly incorporating $\beta$ ~~explicit in~~ into the proximal step. ~~We further show~~ This approximation can also be derived from an approximate KL-projection $\hat{\Gamma}_{\mathbb{U}}^{\mathrm{kl}}(\overset{\circ}{\exp}(-\beta C))$, a typical instance of which is early termination of Sinkhorn scaling. Furthermore, we demonstrate that $\hat{P}_\beta$ can be refined into the exact solution $P_\beta$ ~~via the~~ through an exact KL-projection.

**Proximal point methods** are iterative algorithms for convex optimization, well-suited for nonsmooth or composite objectives. Their core principle involves solving a regularized subproblem at each step. This process optimizes the objective while ensuring the stability of the iterative sequence. To solve the OT (without the entropic term), the proximal point method reads as follows

$$P^{(t+1)} = \arg\min_{P \in \mathbb{U}} C \odot P + \beta^{(t+1)} \mathcal{B}(P, P^{(t)}), \tag{5}$$

$$\mathcal{B}(P, P^{(t)}) = P \cdot \overset{\circ}{\log}(P \oslash P^{(t)}) - (P - P^{(t)}) \cdot \mathbf{1}\mathbf{1}^\top, \tag{6}$$

where $\odot$ is the element-wise product. $\mathcal{B}(\cdot, \cdot)$ is the Bregman divergence used to measure the distance between two probability vectors/matrices (Benamou et al., 2015). This formula can be equivalently rewritten as ~~:~~

$$P^{(t+1)} = \arg\min_{P \in \mathbb{U}} C^{(t)} \odot P + \beta^{(t+1)} \mathcal{H}(P), \quad C^{(t)} = C - \beta^{(t+1)} \overset{\circ}{\log} P^{(t)}. \tag{7}$$

The update can be computed by applying a KL-projection to a modified kernel matrix, yielding:

$$P^{(t+1)} = \Gamma_{\mathbb{U}}^{\mathrm{kl}}\left(\overset{\circ}{\exp}(-C\beta^{(t+1)}) \odot P^{(t)}\right). \tag{8}$$

As $t \to \infty$, $P^{(t)}$ converges to the optimal transport plan. To improve efficiency, Xie et al. (2020) ~~propose~~ proposed the inexact proximal point method (IPOT), which performs only one Sinkhorn iteration to approximate the projection in (8). The resulting sequence $\{\hat{P}^{(t)}\}_{t=1,2,\dots}$ can efficiently converge to the optimal transport plan.

**Inexact proximal point method for entropic OT**. Theorem 1 allows the inexact proximal point method to efficiently offer a $\hat{P}_\beta$ ~~for the entropic optimal transport to compute $P_\beta$. The~~ $\hat{P}_\beta$. This

theorem relies on explicitly tracking the regularization parameter $\beta$ across proximal iterations, as detailed below.

**Lemma 1 (Explicit form of $\beta$)** *The iterative schemes of the exact/inexact proximal point method for entropic OT admit the following explicit representation:*

$$P^{(t)} = \Gamma_{\mathbb{U}}^{\mathrm{kl}}\!\left(\bigodot_{i=1}^{t} \overset{\circ}{\exp}(-C\beta^{(i)})\right), \quad \hat{P}^{(t)} = \hat{\Gamma}_{\mathbb{U}}^{\mathrm{kl}}\!\left(\bigodot_{i=1}^{t} \overset{\circ}{\exp}(-C\beta^{(i)})\right). \tag{9}$$

$\bigodot$ represents element-wise multiplication of a series of matrices. In particular, when each $\beta^{(i)}$ is fixed as $\Delta\beta$, the regularization $\beta$ satisfies $\beta = t\Delta\beta$, which yields ~~a~~ the following explicit expression:

$$\hat{P}^{(t)} = \hat{\Gamma}_{\mathbb{U}}^{\mathrm{kl}}\!\left(\overset{\circ}{\exp}\left(-t\Delta\beta C\right)\right) = \hat{P}_{\beta}, \quad \beta = t\Delta\beta. \tag{10}$$

~~where $\hat{\Gamma}_{\mathbb{U}}^{\mathrm{kl}}$ represents an approximate KL-projection~~ The $\hat{P}_{\beta}$ can admit the form $\hat{P}_{\beta} = D_{(\boldsymbol{u})}\,\overset{\circ}{\exp}(-\beta C)\,D_{(\boldsymbol{v})}$ and satisfies

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

which is obtained by substituting $P$ with $\mathring{\exp}\left(\beta(-C + \boldsymbol{x}\mathbf{1}^\top + \mathbf{1}\boldsymbol{y}^\top) - 1\right),$ to get the solution of (1):

$$f(\boldsymbol{z}) = f(\boldsymbol{x}, \boldsymbol{y}) = -\min_{P \in \mathbb{U}} \mathcal{L}(P, \boldsymbol{x}, \boldsymbol{y}), \tag{14}$$

The explicit form of $f(\boldsymbol{z})$ follows by substituting $P = \mathring{\exp}\left(\beta(-C + \boldsymbol{x}\mathbf{1}^\top + \mathbf{1}\boldsymbol{y}^\top) - 1\right)$ into (13). The corresponding gradient and Hessian matrix matrices of $f$ are

$$\nabla_{\boldsymbol{z}} f(\boldsymbol{z}) = (\nabla_{\boldsymbol{x}} f(\boldsymbol{x}, \boldsymbol{y}), \nabla_{\boldsymbol{y}} f(\boldsymbol{x}, \boldsymbol{y}))) = \left(P\mathbf{1} - \boldsymbol{a}, P^\top \mathbf{1} - \boldsymbol{b}\right),$$

$$H(\boldsymbol{z}) = \nabla^2 f(\boldsymbol{z}) = \nabla^2 f(\boldsymbol{x}, \boldsymbol{y}) = \beta \begin{bmatrix} \text{diag}(P\mathbf{1}) & P \\ P^\top & \text{diag}(P^\top \mathbf{1}) \end{bmatrix}. \tag{15}$$

If $H(\boldsymbol{z}_k)$ is invertible, the Newton method updates $\boldsymbol{z}_{k+1}$ via the formula

$$\boldsymbol{z}_{k+1} = \boldsymbol{z}_k + \Delta \boldsymbol{z}_k, \quad \Delta \boldsymbol{z}_k = H(\boldsymbol{z}_k)^{-1}\left(-\nabla_{\boldsymbol{z}} f(\boldsymbol{z}_k)\right). \tag{16}$$

To avoid costly matrix inversion, $\Delta \boldsymbol{z}$ is obtained by solving a linear system $H\Delta \boldsymbol{z} = -\nabla f$, which is commonly solved by the Conjugate Gradient (CG) method (Tang et al., 2024). However, a key challenge arises from the $\mathcal{O}(n^3)$ computational cost per Newton iteration and the non-invertibility of the Hessian matrix $H$.

### 4.1 FAST POSITIVE-DEFINITE-PRESERVING SPARSE SCHEME

Prior research has made substantial progress in characterizing complexity and irreversibility by sparsifying the Hessian. Tang et al. (2024) observed that, after some Sinkhorn iterations, the Hessian matrix can be well-approximated by a sparse structure. Building on this observation, they reduced the computational complexity of the single Newton update (16) from $\mathcal{O}(n^3)$ to $\mathcal{O}(n^2)$ by enforcing a predetermined sparsity pattern on the Hessian matrix. Building on this Following this direction, Tang and Qiu (2024) proposed an improved sparsification strategy that ensures the invertibility of the sparsified Hessian matrix. However, both approaches suffer from a significant increasing increase in computation time on certain large-scale problems. To overcome this limitation, we propose an efficient sparsification scheme that grantees guarantees an invertible sparse approximation of the Hessian matrix $H$.

The sparsified matrix $\hat{H}$ is constructed by enforcing sparsity on the anti-diagonal submatrices $P$ and $P^\top$:

$$\hat{H} = \beta \begin{bmatrix} \text{diag}(P\mathbf{1}) & \hat{P} \\ \hat{P}^\top & \text{diag}(P^\top \mathbf{1}) \end{bmatrix}. \tag{17}$$

The sparsification procedure comprises the following steps: **Adaptive thresholding:** We truncate the matrices $P$ and $P^\top$ in $H$ by zeroing out all elements with values smaller than a threshold

$\tau = \|\nabla f\|_1 = \rho,$ two steps:

$$\widehat{P}^0_{ij} = \begin{cases} P_{ij}, & P_{ij} \geq \tau, \\ 0, & P_{ij} < \tau, \end{cases} \qquad \widehat{P}_{ij} = \begin{cases} 0, & j = \arg\min_{k: \widehat{P}^0_{ik} > 0} \widehat{P}^0_{ik}, \ (P - \widehat{P}^0)_{i\cdot} = \mathbf{0}, \\ 0, & i = \arg\min_{k: \widehat{P}^0_{kj} > 0} \widehat{P}^0_{kj}, \ (P - \widehat{P}^0)_{\cdot j} = \mathbf{0}, \\ \widehat{P}^0_{ij}, & \text{otherwise.} \end{cases} \qquad (18)$$

where $\rho$ is the marginal error, while capping $\tau$ at $10^{-4}$ $\tau = \|\nabla f\|_1$ is guided by the objective of achieving quadratic convergence (detailed in the proof of Theorem 2). The second step, diagonal dominating operation, ensures that all rows of $\hat{P}$ and $\hat{P}^\top$ contain truncated elements, which also implies that $\hat{H}$ exhibits diagonal dominance and positive definiteness. Appendix H presents the relationship between the sparsity level and $\beta$, and describes how to avoid excessive ~~sparsity (discussed in Appendix H)~~ sparsification which may slow down the optimization.

**Diagonal dominance enforcement:** ~~If every row and column of the sparsified $P$ contains truncated elements, the resulting~~ Excessive sparsification, which removes important entries from the Hessian, may slow down the optimization. To avoid this, we reduce $\tau$ whenever the number of nonzeros in $\hat{P}$ falls below $2n$, ensuring that $\hat{P}$ retains at least $2n$ entries. The choice of $2n$ is motivated by the fact that the solution of the corresponding OT contains at most $2n - 1$ nonzero entries. Thus, if $\hat{P}$ contains more than $2n$ nonzeros, the corresponding $\hat{H}$ ~~is positive definite and hence invertible. Otherwise, we subtract the smallest entry from those rows and columns, ensuring diagonal dominance of symmetric $\hat{H}$, thereby guaranteeing its positive definiteness. The concise design, together with its $\mathcal{O}(n^2)$ complexity, contributes to the remarkable scalability; the positive definiteness of $\hat{H}$ enables stable and efficient CG-based Newton updates; adaptive threshold contributes to local quadratic convergence of the algorithm~~

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

_2^2} \left\| \nabla f\left(\boldsymbol{z}_k\right) \right\|^2 \leq \frac{c_3}{2} \left\| \boldsymbol{z}_k - \boldsymbol{z}^* \right\|^2 \leq \left| f\left(\boldsymbol{z}_k\right) - f\left(\boldsymbol{z}^*\right) \right| \to \epsilon. \tag{103}$$

*Since our line search method and Sinkhorn iteration both satisfy Wolfe condition, applying Lemma 4 we get that $\epsilon = 0$. Combine (102) and (103) with squeeze theorem, we have*

$$\lim_{k \to \infty} \left\| \boldsymbol{z}_k - \boldsymbol{z}^* \right\| = 0. \tag{104}$$

**Theorem 4** *Algorithm 3 converges globally to the solution of the corresponding entropic OT.*

**Proof** *The global convergence of the PSN algorithm follows from the global convergence of its two components: the Proximal-Sinkhorn (PS) and Proximal-Newton (PN). The convergence of PN is established by Theorem 3, while that of PS is guaranteed by the global convergence of the Sinkhorn algorithm Cuturi (2013). Hence, PSN is globally convergent.*

## E    THE SPECTRAL DISTRIBUTION OF THE HESSIAN MATRIX

The following Lemma and Theorem characterize the convergence behavior of CG. Lemma 5 shows that the error at each iteration of CG is bounded by a polynomial $q_k(\lambda)$, and Theorem 5 reveals the connection between the polynomial $q_k(\lambda)$ and the spectral distribution of the Hessian matrix $H$ while solving $H\Delta\boldsymbol{z} = -\nabla f$. ~~Futhermore~~Furthermore, Theorem 5 also demonstrates that the error vector in each iteration of the CG method remains bounded by the polynomial $q_k(\lambda)$, even after excluding a small number of eigenvalues at one or both extremes. This reveals that the CG method's convergence rate is closely tied to the eigenvalue distribution of the matrix $H$, potentially leading to superlinear rates.

**Lemma 5 ((Saad, 2003))** *The approximate solution $\boldsymbol{x}_k$ generated by the CG method of solving $H\boldsymbol{x} = \boldsymbol{b}$ at step $k$ satisfies*

$$\left\| \boldsymbol{x}_k - \boldsymbol{x}^* \right\|_H \leqslant \min_{q_k \in \mathcal{P}_k^{(0)}} \max_{\lambda \in \lambda(H)} \left| q_k(\lambda) \right| \left\| \boldsymbol{x}_0 - \boldsymbol{x}^* \right\|_H, \tag{105}$$

*where $\boldsymbol{x}^*$ denotes the exact solution of the linear system, $\|\cdot\|_H$ represents the $H$-norm, $\mathcal{P}_k^{(0)}$ represents the set of all real coefficient polynomials of degree not exceeding $k$ that satisfy $q_k(0) = 1$.*

**Theorem 5 ((Hestenes et al., 1952))** *Considering the error vectors $\boldsymbol{u}_k = \boldsymbol{x}_k - \boldsymbol{x}^*$ defined by Lemma 5, we establish the following bounds:*

*(1) for $\lambda(H) \subset \{\lambda_1, \cdots, \lambda_p\} \cup [a_1, b_1]$, where $\lambda_i \in \lambda(H), \lambda_i < a_1$, we have*

$$\left\| \boldsymbol{u}_k \right\|_H \leqslant F\left(a_1, b_1; k-p\right) \prod_{i=1}^{p} \frac{b_1}{\lambda_i} \left\| \boldsymbol{u}_0 \right\|_H; \tag{106}$$

*(2) for $\lambda(H) \subset [a_2, b_2] \cup \{\lambda_1', \cdots, \lambda_p'\}$, where $\lambda_i' \in \lambda(H), \lambda_i' > b_2$, we have*

$$\left\| \boldsymbol{u}_k \right\|_H \leqslant F\left(a_2, b_2; k-p\right) \left\| \boldsymbol{u}_0 \right\|_H; \tag{107}$$

*(3) for $\lambda(H) \subset \{\lambda_1, \cdots, \lambda_p\} \cup [a_3, b_3] \cup \{\lambda_1', \cdots, \lambda_p'\}$, where $\lambda_i, \lambda_i' \in \lambda(H), \lambda_i' > b_3, \lambda_i < a_3$, we have*

$$\left\| \boldsymbol{u}_k \right\|_H \leqslant \frac{1}{4} \underset{i=1}{\overset{p}{\prod}} \frac{\lambda_i'}{\lambda_i} \left(1 - \frac{\lambda_i}{\lambda_i'}\right)^2 F\left(a_3, b_3; k-2\right) \left\| \boldsymbol{u}_0 \right\|_H, \tag{108}$$

*where $F(a, b; k) = \left(T_k\left(\frac{b+a}{b-a}\right)\right)^{-1}$, and $T_k(x)$ denotes Chebyshev polynomial of degree $k$.*

## F    PERFORMANCE ON DIFFERENT ENTROPIC PARAMETERS

We evaluate PSN, SNS, SSNS, and the Newton method across entropic parameters $\beta \in \{100, 500, 3000\}$ covering both low and high regularization. Figure 6 indicates that our proposed algorithm, PSN, exhibits superior robustness and effectiveness when $\beta$ is large. For smaller values of $\beta$, PSN reduces to a combination of the proximal point method and Sinkhorn ~~iterations~~scaling, and outperforms other Newton-based algorithms as well as the log-Sinkhorn method.

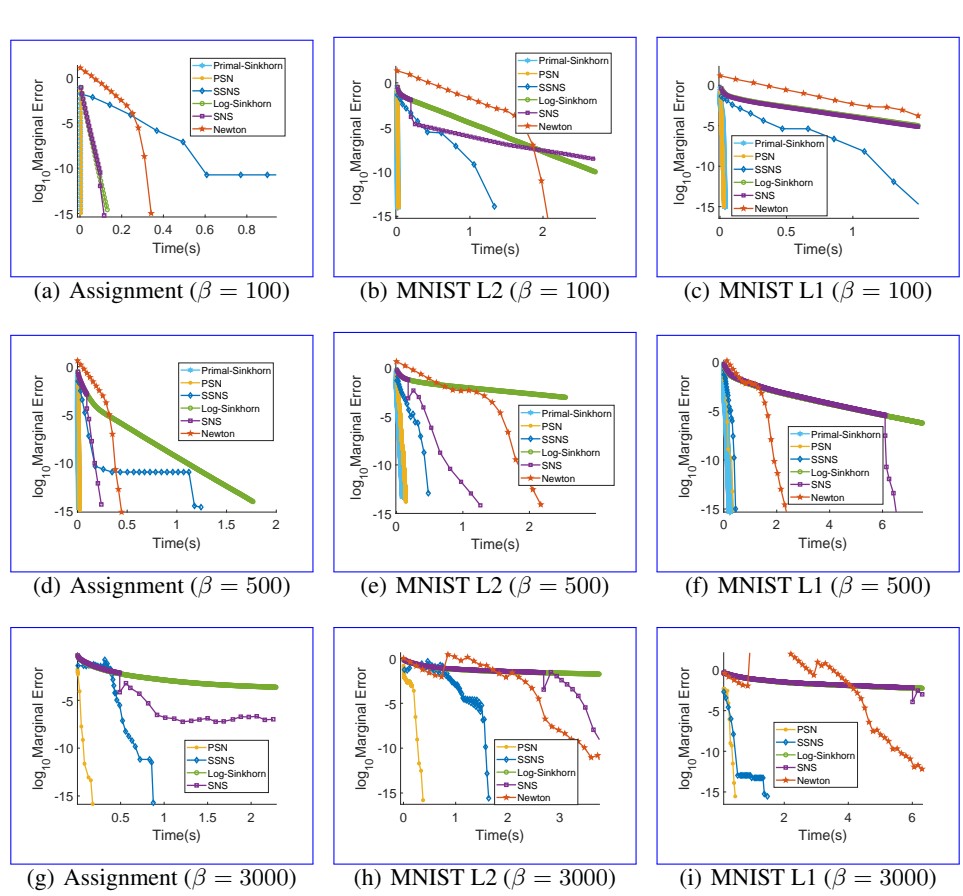

Figure 6: Performance on different $\beta$.

# G COMPARISON WITH OTHER CLASSIC FIRST ORDER ALGORITHMS

In this section, we compare our algorithm against several well-established first order optimization methods, including Limited-memory Broyden–Fletcher–Goldfarb–Shanno algorithm (L-BFGS, (Liu and Nocedal, 1989)) for the quasi-Newton method, the adaptive primal-dual accelerated gradient descent method (ADPAGD, (Dvurechensky et al., 2018)) as an accelerated first-order approach, and the Sinkhorn method implemented as a block coordinate descent method (BCD, (Luo and Tseng, 1992)). We adopted the C++ implementation by Tang and Qiu (2024) for the baseline algorithms[4].

Evaluated across our three datasets, our algorithm, as depicted in Figure 7, consistently delivers superior performance, significantly outperforming these classic optimization techniques in all situations.

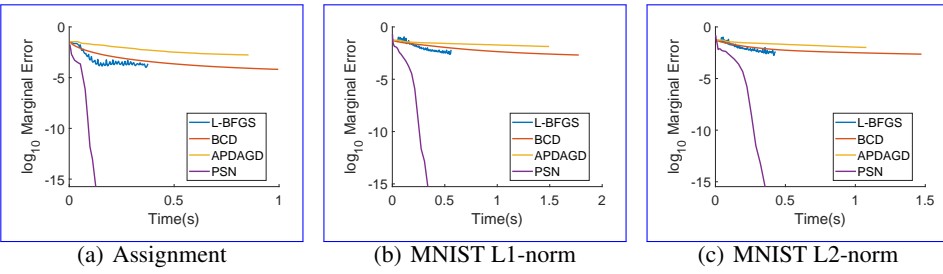

(a) Assignment      (b) MNIST L1-norm      (c) MNIST L2-norm

Figure 7: Other classic algorithms.

# H TRUNCATION DETAILS

We explore the sparsity of the Hessian matrix under various truncation thresholds and visualize the results in Figure 8.

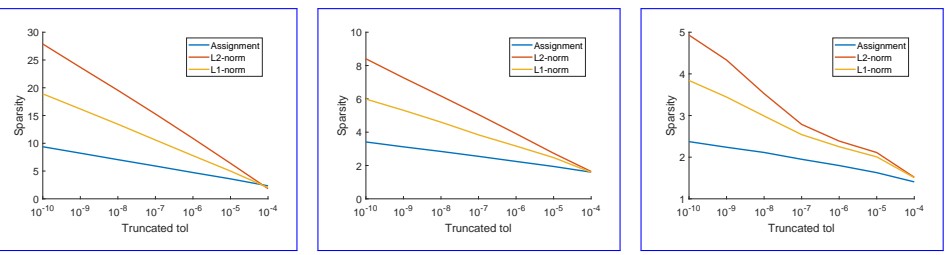

Figure 8: Relationship between truncation threshold and Sparsity. The three subfigures correspond to $\beta = \{1000, 2000, 4000\}$, respectively. The $x$-axis denotes the truncation threshold, while the $y$-axis represents the Hessian matrix sparsity as a multiple of $1/n$.

Figure 8 demonstrates that the truncated matrix exhibits significant sparsity. ~~The number of non-zero elements after applying the $10^{-4}$ threshold is comparable to that of the optimal OT solution, suggesting that a larger truncation threshold may lead to excessive sparsification~~