# OpenReview forum: "A Proximal-Sinkhorn-Newton Method for Entropic Optimal Transport"
_ICLR.cc/2026/Conference — ICLR 2026 Conference Withdrawn Submission_

### Official Review · Reviewer_MkrW · 2025-10-26

**Soundness:** 2
**Presentation:** 3
**Contribution:** 2
**Rating:** 2
**Confidence:** 5

**Summary:**

This paper introduces a hybrid algorithm, the Proximal-Sinkhorn-Newton (PSN) method, combining an inexact proximal point step for stability with a sparse Newton refinement for efficiency in solving entropic optimal transport (EOT) problems. The authors establish local quadratic and global convergence results and report empirical speedups over existing Sinkhorn and Newton-type solvers.

While the paper is well structured and presents a coherent algorithmic framework, its technical and empirical contributions are incremental relative to existing Newton-based OT solvers such as “A Truncated Newton Method for Optimal Transport” (Kemertas et al., 2025).

**Strengths:**

(1) A clear integration of proximal regularization with sparse Newton iterations and with theoretical justification, including convergence proofs and use of incomplete Cholesky preconditioning.
(2) A unified “inexact-to-exact” interpretation that is conceptually appealing.

**Weaknesses:**

(1) Limited test scale. Numerical tests are not compelling—datasets have at most n = 5000, far below what recent OT solvers handle. To support the claimed “20× speedup,” larger-scale experiments (≥ 10⁵ samples) or GPU benchmarks would be necessary.
(2) Analytical specificity. The spectral estimates (e.g., Eqs. (54)–(56), (80)) largely follows standard proofs. The claimed local quadratic rate and global convergence are textbook results; a more informative contribution would involve explicit spectral estimates for the Hessian H in terms of sample size n, feature dimension d, and the EOT parameter \beta.
(3) Complexity discussion. The runtime analysis is qualitative. It would strengthen the paper to derive complexity bounds tied to the annealing schedule of \Delta \beta, and to compare analytically (or experimentally) how different \Delta\beta choices affect convergence.
(3) Comparison depth. The comparison to prior Newton-type methods (e.g., SNS, SSNS, Truncated Newton) lacks sufficient quantitative context—please include wall-clock and iteration counts over a wider range of \beta and n.

**Questions:**

(1) I wonder if the authors can provide more explicit spectral estimates for the Hessian H in terms of sample size n, feature dimension d, and the EOT parameter \beta, at least numerically.
(2) I wonder if the authors can derive complexity bounds tied to the annealing schedule of \Delta \beta, and to compare analytically (or numerically) how different \Delta\beta choices affect convergence.
(3) The comparison to prior Newton-type methods (e.g., SNS, SSNS, Truncated Newton) lacks sufficient quantitative context—please include more comparison such as the wall-clock and iteration counts over a wider range of \beta and n.

---

> ### Author Response · Authors · 2025-11-22
> **Response to Reviewer MkrW**
>
> We thank the Reviewer for their positive assessment of our paper’s structure and “coherent algorithmic framework.” We also appreciate the helpful reference (Kemertas et al., 2025), which we have included in the revised manuscript to better situate our contribution within the broader literature.
>
> We would like to clarify the core conceptual contribution of our method. The proposed Proximal-Sinkhorn-Newton (PSN) framework is not strictly limited to an inexact proximal point method + Newton method combination. More fundamentally, it is built around the general structure of an inexact proximal point method + KL projection. This KL projection step can be realized by a variety of solvers: the Sinkhorn scaling, quasi-Newton methods or even the truncated Newton method mentioned by the Reviewer. This design makes our framework more general and flexible than a fixed Newton-type solver.
>
>
> More fundamentally, our Lemma 1 characterizes the associated regularization parameter $\beta$ of proximal point methods, including annealing schemes such as annealed Sinkhorn and mirror Sinkhorn, at each step. This perspective enables this kind of OT algorithms to be effectively applied to entropic OT problems. The truncated newton method suggested by the Reviewer may also benefit from such a unified view.
>
>  We hope this clarification will help the reviewer better understand the generality and extensibility of our approach.
>
> ---
>
>
> We thank the reviewer for the helpful suggestions, which have been incorporated into the revised version. We have revised the manuscript in accordance with all of your suggestions, and we have included a comparison between the previous and revised versions in the supplementary material.
>
>
> **W1. (Limited test scale)**
> Regarding the “20× speedup” mentioned, we would like to clarify that the original text specifically stated this was observed “on some problems” (Table 4), within the context of specific entropic OT instances. To avoid any confusion, we have removed this phrasing from the revised manuscript.
>
> We thank the reviewer for raising this important point. In response, we have added additional experiments (\(n = 10,000\)) in Appendix R. We have not tested larger problem sizes because \(n = 10, 000\) is close to the memory limit of our current CPU environment. We are working on a GPU-based PyTorch implementation to enable larger-scale experiments, but since PyTorch lacks built-in IC preconditioning and CG modules, this requires substantial additional development and is still in progress.
>
>
>
> **W2\&Q1. (Analytical specificity)**
> Thank you for this insightful comment.
>
> (1) On explicit spectral estimates: We completely agree with the reviewer that obtaining explicit spectral estimates for the Hessian H would be a valuable contribution.  In response, we have given a theoretical analysis and added numerical experiments in Appendix Q of the revised manuscript.
>
> (2) On the novelty of convergence analysis: Regarding the point that the local quadratic rate and global convergence are "textbook results," we would like to respectfully clarify that our contribution is not a textbook result but a direct consequence of our new design. The convergence is precisely our use of Sinkhorn iterations to escape plateauing that enables the global convergence property. Our new adaptive thresholding contributes to the quadratic property. Our baselines (SNS and Newton) do not have such two properties.
>
>
> **W3&Q3. (Comparison depth)**
> We thank the reviewer for the suggestion to broaden the quantitative comparison. We agree that a wider range of the entropic parameter $\beta$ would be beneficial for a comprehensive evaluation. Our current study have significantly expanded the experimental scale compared to prior Newton-type works in entropic OT:
> - Our Work:  $\beta \in (100, 3000)$.
> - SSNS: $\beta =100$  or  $1000$.
> - SNS: $\beta =1200$.
>
> Therefore, we believe our current experimental framework, which already surpasses the scope of prior works in $\beta$. Besides, the truncated Newton mentioned by the reviewer did evaluate algorithms on different $\beta$ since their problem is OT not entropic OT. We sincerely hope this clarification addresses the reviewer's concern and are happy to provide further analysis on specific parameter ranges if needed.
>
> **Q2. ($\Delta\beta$ choices)**
> Thanks for this question. We have studied the sensitivity of PSN with respect to the proximal step size $\Delta\beta$ in Appendix M of the revision. Since for a given $\beta$, the relation
> $$
> \ell = \left\lceil \frac{\beta}{\Delta \hat{\beta}} \right\rceil
> $$
> holds, results in Sec. 6.1 also implicitly demonstrate the sensitivity of IP-EOT to the $\Delta \beta$. Specifically, a smaller $\Delta\beta$ (equivalently, a larger $\ell$) leads to a smaller final marginal error.

---

> > ### Comment · Reviewer_MkrW · 2025-11-22
> >
> > I would like to thank the authors for the revision. While I agree that the new design and adaptive thresholding improve the convergence of EOT, I am not fully convinced by the numerical experiments, as the dataset size is not comparable to the existing methods.

---

> > > ### Author Response · Authors · 2025-11-28
> > > **Response to Reviewer MkrW (Second Round)**
> > >
> > > We sincerely thank the reviewer for their positive feedback. After nearly a week of intensive work, we carried out the $10^5$ scale experiments as suggested. To address memory limitations, we adopted a non-explicit storage strategy, computing the required matrices dynamically during execution. This allowed us to implement an $\mathcal{O}(n)$-memory version of our method in CPU-based MATLAB. Although reducing the memory footprint inevitably increases the computational time of IP-EOT, the results clearly demonstrate the scalability of our algorithm. For the reviewer’s convenience, a summary of the experimental results is provided in the table below, and more detailed results can be found in Appendix R of the revised manuscript.
> > >
> > > | Marginal error | $10\^{-2}$ | $10\^{-4}$ |$10\^{-6}$ | $10\^{-8}$ | $10\^{-12}$ | $10\^{-14}$ |
> > > |----------------|-------------|-------------|-------------|-------------|--------------|--------------|
> > > | Time           | 799s        | 18321s      | 20361s      | 20727s      | 21115s       | 21526s       |
> > >
> > > We hope that these results can address the reviewer’s concerns.

---

### Official Review · Reviewer_YPFa · 2025-11-01

**Soundness:** 2
**Presentation:** 3
**Contribution:** 2
**Rating:** 4
**Confidence:** 3

**Summary:**

The paper is dedicated to efficient computation of entropic optimal transport (OT) between discrete distributions. Achieving high precision with low regularization poses a major challenge for existing algorithms. For example, the famous Sinkhorn algorithm becomes unstable for such formulations. Proximal point methods relax these problems into a series of subproblems with stronger regularization, which improves stability but requires careful tuning of the regularization parameters. Newton methods require proper initialization and have a prohibitively large iteration cost, while their sparse variants suffer from certain stagnation issues. To overcome these difficulties, the authors propose versions of inexact proximal point and sparse Newton methods, as well as a hybrid solver. The experiments showcase the fast convergence of the proposed methods.

**Strengths:**

1. The paper presents a theoretical result on the proximal point method for entropic OT, providing a simple way to choose the regularization parameters for its subproblems; this provides a practical guideline for parameter selection.
2. The authors propose a new efficient Hessian sparsification procedure tailored to OT problems. Two convergence theorems for the related procedures are provided.
3. The experiments demonstrate that the proposed hybrid method is efficient across several different scenarios.

**Weaknesses:**

1. In some parts of the paper, the presentation is informal. For example, the statement of Lemma 2 does not specify what is meant by the word "approximation". Furthermore, the fast positive-definite-preserving sparse scheme should be formally defined. Theorems 2 and 3 should clearly state the assumptions on the parameters of the procedures. The term "objective score" should be defined.
2. The hybrid method lacks convergence guarantees.
3. The paper contains a lot of typos and small errors.

**Questions:**

1. Where appropriate, could you perform multiple runs of experiments with different realizations of randomness, and depict the variance in the plots?
2. Is it possible to formulate some convergence statement regarding Algorithm 3?
3. Could you please provide a zoom-in for the plots where some curves almost overlap, e.g., Figure 5(a).

Typos and minor errors:
- Line 107: "it can serve**s**"
- Theorem 1 (and some of the other statements): "Let ... denote**s**"
- Equations (5), (7): did you accidentally use elementwise product instead of the Frobenius inner product? Also, please introduce the notation for the elementwise product before using it.
- Equations (9): notation for an approximate KL-projection is used here, but introduced only after equation (10)
- Algorithm 2, line 5: is PCG defined in the text?
- Line 279: "grantees"
- Line 313: hat should be above H
- Line 323: missing space
- Line 363: "entries **[are?]** sampled"
- Line 380: missing space

---

> ### Author Response · Authors · 2025-11-22
> **Response to Reviewer YPFa**
>
> We extend our sincere thanks to the reviewer for their generous feedback. Their acknowledgment of the practical utility of our theoretical findings, the novelty of our Hessian procedure, and the effectiveness of our hybrid method means a great deal to us. We are thrilled that our work has been received so positively and appreciate their supportive perspective.
>
>
> We thank the reviewer for the helpful suggestions, which have been incorporated into the revised version. We have revised the manuscript in accordance with all of your suggestions, and we have included a comparison between the previous and revised versions in the supplementary material.
>
>
> **W1. (Informal presentation)**
> We thank the reviewer for pointing out these presentation issues.
>
> We agree that the "approximation" should be stated precisely. In the revised manuscript, we have provided a formal definition of the approximation before Lemma 2.
>
> Second, the fast positive-definite-preserving sparse scheme has been formally and rigorously defined in the revision. This includes explicitly specifying the construction of the sparsified Hessian.
>
> Regarding the assumptions in Theorems 2 and 3:
>  Theorem 2 concerns the sparse Newton step, and the only parameter involved is the sparsification threshold $\tau$, which has already been introduced and formally defined in the fast positive-definite-preserving sparse scheme. No additional assumptions on $\tau$ are required for the theorem to hold.
>  Theorem 3 analyzes Algorithm 2. The parameters appearing in this algorithm are $\beta$ and $\rho\_{\mathrm{th}}$. The value of $\beta$ is provided by the user, and $\rho\_{\mathrm{th}}$ serves solely as a stopping criterion. Importantly, neither parameter affects the convergence guarantees, and therefore no further assumptions are necessary.
> In summary, neither theorem requires any parameter assumptions beyond those already specified in the problem (e.g., $\beta \neq 0$).  If the reviewer has more specific suggestions, we are fully willing to further improve the manuscript.
>
> Finally, the term *objective score* have been replaced by  *objective value*  and properly defined. Specifically, it refers to the objective value $f(z)$.
>
> We appreciate the reviewer’s helpful suggestions and believe these revisions will significantly improve the clarity and rigor of the presentation.
>
> **W2&Q2.  (Convergence of hybrid method)**
> We appreciate the reviewer for pointing out this omission. In the revised manuscript, we have included a formal convergence guarantee for the hybrid method in section 5, together with a complete proof to address the reviewer’s concern.
>
> **W3.  (Typos and small errors)**
> We thank the reviewer for the careful reading. In the revised manuscript, we have thoroughly proofread the entire text and correct all such issues to ensure clarity and accuracy.
>
> **Q1&3. (Variance and zoomed-in views in plots )**
> We thank the reviewer for the helpful suggestions.
>
> We agree that reporting results over multiple runs and illustrating the variance would improve the clarity of the empirical evaluation. In Appendix N of the revised version, we have performed multiple independent runs for the relevant experiments and include variance in the corresponding plots.
>
> Regarding the visualization issue, we also agree that some curves in the current figures overlap closely (e.g., Figure 5(a)). We have provided zoomed-in views in Appendix O of the revised manuscript to allow for a clearer comparison among the methods. This material is placed in the appendix temporarily to facilitate the review process and will be incorporated into the main text in the final version.
>
> We believe these additions will significantly enhance the readability and completeness of the experimental section.

---

### Official Review · Reviewer_QcbC · 2025-11-02

**Soundness:** 3
**Presentation:** 3
**Contribution:** 3
**Rating:** 6
**Confidence:** 4

**Summary:**

The paper proposes the Proximal-Sinkhorn-Newton (PSN) method, a hybrid algorithm for solving entropic optimal transport problems. The method combines an inexact proximal point method (IP-EOT) to generate a stable initial solution, which is then refined by either Sinkhorn iterations or a sparse Newton method. The authors provide theoretical analysis for their framework and present experimental results showing that PSN is faster than several baseline methods on the tested problems.
The paper makes a solid contribution by demonstrating how three well-known optimization techniques can be integrated to address the practical challenges of entropic OT. The approach is logical and the empirical results are positive. However, the theoretical guarantees follow standard forms and have practical limitations, and the introduction of new hyperparameters adds to tuning complexity. On balance, the paper's contributions warrant acceptance.

**Strengths:**

1. A key contribution is Theorem 1, which formalizes the connection between the iterates of the IP-EOT method and the entropic OT solution for a linearly increasing regularization parameter β. This provides a clear justification for using the IP-EOT output as a warm-start for the refinement stage.
2. The paper's main strength is the design of a multi-stage algorithm that leverages the advantages of different optimization methods. Using a proximal point method to ensure numerical stability before switching to a faster second-order method is a sound and logical strategy. The adaptive switch between Sinkhorn and Newton based on Hessian sparsity is a thoughtful addition for improving robustness.

**Weaknesses:**

1. The framework introduces new, important hyperparameters, namely the proximal step-size $\Delta \beta$ and the sparsity threshold $\lambda$. The performance of the algorithm appears to depend on these settings, but the paper lacks a sensitivity analysis or ablation study. This makes it difficult for a practitioner to understand how to tune these parameters for new problems, and whether the reported performance is robust to these choices.
2. The global convergence proof establishes asymptotic convergence but does not characterize the constants that govern the actual speed.

**Questions:**

1. Could you provide some intuition on how a user should set the $\Delta \beta$ and $\lambda$  parameters?

---

> ### Author Response · Authors · 2025-11-22
> **Response to Reviewer QcbC**
>
> We extend our sincere thanks to the reviewer for their generous and thoughtful comments. Their recognition of the theoretical justification provided by Theorem 1 and the logical, robust architecture of our multi-stage algorithm means a great deal to us. We are delighted that the core ideas behind our work resonate so well, and we appreciate their supportive perspective.
>
>
> We thank the reviewer for the helpful suggestions, which have been incorporated into the revised version. We have revised the manuscript in accordance with all of your suggestions, and we have included a comparison between the previous and revised versions in the supplementary material.
>
>
>
> **W1&Q1. (Hyperparameters)**
> We thank the reviewer for the insightful comments.
>
> **Proximal step size** $\Delta \beta$.
> We have studied the sensitivity of PSN with respect to the proximal step size $\Delta\beta$, accompanied by additional experiments in Appendix M. Since for a given $\beta$, the relation
> $$
> \ell = \left\lceil \frac{\beta}{\Delta \hat{\beta}} \right\rceil
> $$
> holds, these results in Section 6.1 also implicitly demonstrate the sensitivity of IP-EOT to the proximal step size $\Delta \beta$. Specifically, a smaller $\Delta \beta$ (equivalently, a larger $\ell$) leads to a smaller final marginal error. From the perspective of the entire PSN pipeline, a smaller $\Delta \beta$ yields a more accurate initialization for the second-stage refinement.
>
>
> **Switching parameter** $\lambda$.
> The primary purpose of $\lambda$ is to decide, whether Proximal-Sinkhorn or Proximal-Newton provides the more efficient update. To further justify the robustness of this mechanism, we have added a new subsection in the Appendix J presenting the values of $|\hat{H}|_0$ under different $\beta$, along with a comparison of the computational efficiency between the two update rules in each regime.
>
>
> **W2. (Global convergence rate)**
> Thank you for raising this question. Unfortunately, the classical Newton method does not, in general, possess a global convergence rate. In fact, standard Newton’s method is only guaranteed to achieve local quadratic convergence, and it lacks any form of global convergence guarantee.
>
> In our work, global convergence is ensured not by the Newton step itself, but by our specially designed line-search–based step-size strategy, which allows the algorithm to make progress even when the pure Newton step fails. This mechanism (detailed in Section 4.3 of the revised manuscript) enables us to establish global convergence, while still retaining the fast local convergence behavior of Newton’s method.

---

### Official Review · Reviewer_pAcV · 2025-11-02

**Soundness:** 3
**Presentation:** 3
**Contribution:** 3
**Rating:** 4
**Confidence:** 3

**Summary:**

The paper proposes PSN, a hybrid solver for entropic optimal transport (EOT) that combines (1) a proximal homotopy scheme (IP-EOT), (2) a sparse preconditioned Newton refinement, and (3) an adaptive switching rule. It targets the regime of large $\beta$ (weak regularization), where classical Sinkhorn methods become numerically unstable. Experiments show that the method works well on small scale problems.

**Strengths:**

- The paper directly addresses a known limitation of the Sinkhorn algorithm: numerical instability and slow convergence when $\beta$ is large. By introducing a proximal continuation scheme (IP-EOT, Algorithm 1), the authors ensure stable iterations even under weak regularization. This makes the contribution both timely and practically relevant for modern OT solvers.
- The convergence of the proposed algorithms are provided by the solid theoretical results.
- Experiments on small scale problems show the effectivity.

**Weaknesses:**

- The method’s performance depends on several hyperparameters such as the proximal step size $\Delta \beta$, sparsification threshold $\tau$, and switching parameter $\lambda$ (Sec. 4–5). These are chosen heuristically, with no ablation or sensitivity analysis provided. As a result, it is unclear how robust PSN remains under different settings or across diverse cost structures.
- All reported experiments are conducted on moderate-sized problems (n ≤ 4096), while modern OT applications often involve tens or hundreds of thousands of samples. Without results on larger datasets, the scalability and numerical stability of the proposed solver remain uncertain, especially considering the overhead of Hessian construction and preconditioning.
- The empirical study focuses mainly on synthetic linear assignment and MNIST benchmarks. No results are shown on more complex or real-world machine-learning applications such as domain adaptation, graph matching, or OT-based generative models. This limits the evidence that PSN can generalize beyond controlled test cases.

**Questions:**

See above

---

> ### Author Response · Authors · 2025-11-22
> **Response to Reviewer pAcV**
>
> We thank the reviewer for their positive and encouraging comments. We are very pleased that the core contributions of our work—addressing the numerical instability of Sinkhorn for large entropic regularization and introducing a stable proximal continuation scheme—are recognized as both timely and practically relevant. It is also gratifying that the solidity of our theoretical convergence analysis and the effectiveness of our method in experiments are appreciated. We are grateful for this supportive feedback.
>
>
> We thank the reviewer for the helpful suggestions, which have been incorporated into the revised version. We have revised the manuscript in accordance with all of your suggestions, and we have included a comparison between the previous and revised versions in the supplementary material.
>
>
> **W1. (Hyperparameters)**
>
> **Proximal step size $\Delta \beta$.** We have studies the sensitivity of PSN with respect to the proximal step size $\Delta\beta$ in the Appendix M of revised manuscript, accompanied by additional experiments.  Since for a given $\beta$, the relation
> $$
> \ell = \left\lceil \frac{\beta}{\Delta \hat{\beta}} \right\rceil
> $$
> holds, these results in Section 6.1 also implicitly demonstrate the sensitivity of IP-EOT to the proximal step size $\Delta\beta$. Specifically, a smaller $\Delta\beta$ (equivalently, a larger $\ell$) leads to a smaller final marginal error. From the perspective of the entire PSN pipeline, a smaller $\Delta\beta$ yields a more accurate initialization for the second-stage refinement.
>
> **Sparsification threshold $\tau$**.
> The threshold $\tau$ is in fact not chosen heuristically. Its selection follows directly from the proof of the local quadratic convergence in Theorem 2. We have explicitly clarified this point in the revised manuscript  to avoid potential misunderstanding.
>
> **Switching parameter $\lambda$.**
> The primary purpose of $\lambda$ is to decide, whether Proximal-Sinkhorn or Proximal-Newton provides the more efficient update. To further justify the robustness of this mechanism, we have added a new section in the Appendix J presenting the values of $|\hat{H}|_0$ under different $\beta$, along with a comparison of the computational efficiency between the two update rules in each regime.
>
>
>
> **W2. (Moderate-sized problems)**
> We thank the reviewer for raising this important point regarding scalability. In response, we have now included additional experiments ($n=10000$) in Appendix R of the revised manuscript. We have not yet conducted experiments on larger problem sizes because a dimension of 10,000 is close to the memory limit of our current CPU environment. We are now working on implementing the entire codebase in PyTorch on the GPU in order to perform larger-scale experiments. However, since PyTorch does not provide built-in modules for IC preconditioning and PCG, we need to implement these components manually. This is a substantial undertaking, and the work is still in progress.
>
> Regarding the computational overhead, we would like to clarify that our method constructs a sparse Hessian $H$ by sparsifying the $P$. This procedure results in a Hessian with only $\mathcal{}O(n)$ non-zero entries when $\beta$ is large. Consequently, the costs associated with both Hessian construction and preconditioning scale linearly with the problem size, making them manageable even for larger values of $n$. We hope these additional results and clarifications help to alleviate the concern.
>
> **W3. Real-world application**
> We agree with the reviewer that demonstrating performance on complex tasks is important. To further strengthen our study, we have performed additional experiments on ImageNet, shown in Appendix L.
>
> We also appreciate the opportunity to clarify that entropic OT is essentially used as an approximation for solving linear assignment problems in graph matching, our experiments on linear assignment can be viewed as validating the algorithm's application in this domain. We have revised the manuscript to include this perspective and hope these revisions adequately address the reviewer's concern.

---

> > ### Author Response · Authors · 2025-11-28
> > **Response to Reviewer pAcV (2)**
> >
> > After nearly a week of intensive work, we carried out the $10^5$ scale experiments to evaluate scalability and numerical stability as suggested. To address memory limitations, we adopted a non-explicit storage strategy, computing the required matrices dynamically during execution. This allowed us to implement an $\mathcal{O}(n)$-memory version of our method in CPU-based MATLAB. Although reducing the memory footprint inevitably increases the computational time of IP-EOT, the results clearly demonstrate the scalability of our algorithm. For the reviewer’s convenience, a summary of the experimental results is provided in the table below, and more detailed results can be found in Appendix R of the revised manuscript.
> >
> > | Marginal error | $10\^{-2}$ | $10\^{-4}$ |$10\^{-6}$ | $10\^{-8}$ | $10\^{-12}$ | $10\^{-14}$ |
> > |----------------|-------------|-------------|-------------|-------------|--------------|--------------|
> > | Time           | 799s        | 18321s      | 20361s      | 20727s      | 21115s       | 21526s       |
> >
> > We hope that these results can address the reviewer’s concerns.

---

### Official Review · Reviewer_C7nk · 2025-11-03

**Soundness:** 3
**Presentation:** 3
**Contribution:** 2
**Rating:** 4
**Confidence:** 4

**Summary:**

This paper introduces a novel **Proximal-Sinkhorn-Newton** (PSN) method for entropic optimal transport, designed to address the challenges of numerical instability and slow convergence. The approach employs a two-stage framework that integrates an inexact proximal point method with (optionally) a sparse Newton refinement step.
The algorithm can be summarized as follows:
- For a given regularization parameter $\beta$, construct a sequence of intermediate regularization values that increase linearly up to $\beta$, and obtain an approximate transport plan using the proposed IP-EOT method.
- Based on the characteristics of this approximate solution, refine it by applying either the sparse Newton method or exact Sinkhorn iterations.

**Strengths:**

**Linking the “Inexact” and “Exact” paradigms**: Theorem 1 provided a link between the result of IP-EOT and the exact EOT solution, which may benefit future EOT methods using inexact initialization.

**Verifiable numerical stability and efficiency**: Through numerical experiments, it is demonstrated that PSN can avoid underflow issues common in Sinkhorn under large $\beta$. Without the heavy overhead of log-domain and/or using exact Hessian, PSN can also outperform other 1st-order methods significantly.

**Weaknesses:**

**Limited comparison with existing methods**: Solving entropic OT with varying regularization parameters is not a new concept. For instance, [1] proposed a log-stabilized approach with exponentially scaled $\beta$, which is implemented as `sinkhorn_epsilon_scaling()` in the Python POT package and is easily accessible to practitioners. For a fair evaluation, the authors should include this method in their experiments.

Furthermore, as an ablation study, the authors could consider comparing IP-EOT with other approximate entropic OT methods, such as Screenkhorn [2].

**Lack of discussion on hyperparameters**: The PSN algorithm introduces several hyperparameters, yet the paper provides little guidance on how to select them. It would be helpful to include a discussion addressing questions such as: (i) how sensitive are IP-EOT and PSN to the choice of $\Delta\beta$? (ii) what is the impact of using exponential rather than linear scaling in IP-EOT? (iii) how should the truncation cap $\tau$ be chosen to avoid excessive sparsification? and (iv) how does excessive sparsification (Appendix H) affect the final solution?

**Unclear graph**: Figure 4(a) is intended to illustrate that IC preconditioners produce a more concentrated eigenvalue distribution, yet the plot appears to suggest the opposite. Only upon very close inspection can one notice the red points beneath the blue ones. This detail is easily overlooked and will be lost in print. The authors should use a clearer visualization method, such as histograms, to better convey their message.

**Minor issues/typos**:
- Line 380: missing a space between “log-Sinkhorn” and the reference after.
- Line 986: “Futhermore” should be “Furthermore”.
- Line 1133: “.com” in the footnote comes with an extra “m”.

**References**:

[1]: Schmitzer, B. (2016). Stabilized Sparse Scaling Algorithms for Entropy Regularized Transport Problems. arXiv:1610.06519.

[2]: Alaya M. Z., Bérar M., Gasso G., Rakotomamonjy A. (2019). Screening Sinkhorn Algorithm for Regularized Optimal Transport. arXiv:1906.08540

**Questions:**

The article demonstrates that IC preconditioners can substantially enhance the performance of Newton methods. I am interested in understanding under what conditions can such improvements be expected, and whether there is anything specific about the Lagrange function in equation (12) or the problem setup (e.g. assignment vs $L_1$ vs $L_2$) that facilitates these conditions. I understand these questions may be somewhat broad, so even general intuition or insights would be greatly appreciated.

---

> ### Author Response · Authors · 2025-11-22
> **Response to Reviewer C7nk (Weaknesses part)**
>
> We thank the reviewer for the positive assessment. We appreciate the recognition of our contribution in connecting the Inexact and Exact paradigms (Theorem 1), as well as the verification of PSN’s numerical stability and efficiency compared to Sinkhorn and other first-order methods. Your encouraging comments are sincerely appreciated.
>
> We thank the reviewer for the helpful suggestions, which have been incorporated into the revised version. We have revised the manuscript in accordance with all of your suggestions, and we have included a comparison between the previous and revised versions in the supplementary material.
>
> **W1. (Limited comparison with existing methods)**
> We fully agree that adding additional baselines would strengthen our experimental evaluation. In the revised version, we have included a comparison with the sinkhorn\_epsilon\_scaling() implementation from the POT library, and we provide the results in the Appendix K for your review.
>
> Regarding Screenkhorn, our experiments indicate that it is difficult to achieve marginal errors below $10^{-2}$, which is consistent with the results reported in its original paper.  For this reason, we did not include it as a baseline in our main comparisons.
>
> **W2. (Hyperparameters)**
> **W2(i). ($\Delta \beta$)**  We thank the reviewer for raising this question. We have included additional experiments to analyze the sensitivity of PSN with respect to $\Delta\beta$ in the Appendix M. Regarding IP-EOT, note that the number of iterations $\ell$ and the step size $\Delta\beta$ are inherently coupled through $\ell = \beta / \Delta\beta$. Therefore, the sensitivity of IP-EOT to $\Delta\beta$ can already be observed from the results in Section 6.1.
>
> **W2(ii). (Exponential scaling)**
> This is an interesting question. Before answering, we explicitly define the two scaling schemes as follows.
> **Linear Scaling**
> $$
> \Delta\beta = \frac{{\beta}}{\ell},  \quad  \beta_k =  k  \Delta\beta, \quad k = 0, 1, 2, \ldots, \ell.
> $$
>
> **Exponential Scaling** from initial value $\beta_0$ to target value ${\beta}$:
> $$
> \beta\_k = \beta\_0 \gamma^k, \quad \gamma = \left(\frac{{\beta}}{\beta_0}\right)^{1/\ell}, \quad k = 0, 1, 2, \ldots, \ell
> $$
> $$
> \Delta\beta\_k = \beta\_{k+1} - \beta\_k = \beta\_0 \gamma^k (\gamma - 1)
> $$
> Under exponential scaling, $\Delta\beta_k$ varies at each step, making numerical stability more challenging to control.  Moreover, exponential scaling requires specifying two parameters: the initial value $\beta\_0$ and the number of iterations $\ell$.
>
> **W2(iii\&iv). (Truncation cap and excessive sparsification)**
>
> For (iii), to avoid excessive sparsification, the truncation cap $\tau$ should be chosen such that it does not exceed $10^{-4}$. If $\tau$ is set too large (e.g., above this threshold), the number of non-zero elements after sparsification may fall below $2n$, whereas the unregularized optimal transport solution can contain up to $2n-1$ active elements. Removing too many entries risks discarding important components that are necessary for an accurate transport plan.
>
> Inspired by the reviewer's question, we have refined the method for setting the upper bound of $\tau$ to ensure that $\hat{H}$ retains at least $6n$ non-zero elements. This guarantees that $\hat{P}$ contains at least $2n$ non-zero entries, thereby preventing the truncation of important elements. This approach has been thoroughly described in Section 4.1 and the Appendix of the revised manuscript.
>
> Regarding (iv), excessive sparsification—such as setting $\tau = 10^{-2}$—causes the sparsified Hessian to deviate significantly from the exact Hessian. This deviation results in poor descent directions during Newton iterations and a much slower reduction in marginal error.
>
>
> **W3. (Unclear graph)** We appreciate the reviewer for carefully reading and pointing out this issue. In the revised version of the manuscript, we have replaced Figure 4(a) with clearer visualizations. Specifically, we now present the eigenvalue distributions in separate plots to avoid overlap and ensure that the effect of the IC preconditioners can be easily distinguished.

---

> ### Author Response · Authors · 2025-11-22
> **Response to Reviewer C7nk (Questions part)**
>
> **Q1.  IC preconditioner**
> This is a nice question. The effectiveness of the Incomplete Cholesky (IC) preconditioner is theoretically governed by the following two factors:
>
>     Condition Number: The IC preconditioner cannot fully resolve issues arising from an extremely ill-conditioned system. In such cases, the convergence rate of the iterative solver is likely to remain slow.
>
>     Sparsity: The efficiency of the IC preconditioner is positively correlated with the sparsity of the matrix; the sparser the matrix, the more significant the performance improvement.
>
> In entropic OT, when the elements of the coupling matrix $P$ are too similar (e.g., due to nearly identical entries in the cost matrix or a small entropy regularization coefficient $\beta$), the resulting Hessian matrix $H$ may tend to exhibit near-linear dependencies, leading to a worse condition number. This not only impedes the convergence of Newton's method, but also substantially limits the benefit of applying the IC preconditioner. Moreover, smaller variations among the elements of $P$ generally lead to lower sparsity in $H$, which further undermines the efficiency of the IC preconditioner, as its performance relies heavily on the matrix sparsity structure.
>
> In summary, greater variation among the elements of the cost matrices $C$ (which leads to a similar property in  $P$) and a larger value of $\beta$ are likely to enhance the effectiveness of the IC preconditioner.

---

### Official Review · Reviewer_nCyg · 2025-11-03

**Soundness:** 3
**Presentation:** 3
**Contribution:** 2
**Rating:** 4
**Confidence:** 3

**Summary:**

This paper introduces inexact proximal point methods(IP-EOT) and fast sparse newton method, and its integrated framework proximal sinkhorn newton(PSN) method to target entropic optimal transport(EOT). IP-EOT constructs numerically stable warm starts by increasing the regularization in small increments, mitigating underflow. The sparse Newton phase achieves global convergence via backtracking line search with Sinkhorn fallback, and enjoys fast local (near-quadratic) convergence near the solution. Each method shows low computational complexity ($O(n^2)$ per iteration) compared to other methods for EOT problem.  Empirically, PSN delivers faster and more stable convergence than competing EOT solvers across regularization regimes.

**Strengths:**

1. The paper introduces a proximal scheme for entropic OT with an **Inexact→Exact transition** (Theorem 1), splitting $\beta$ into $l$ small steps to avoid underflow/instability and warm-start refinement.
2. The Fast Sparse Newton combines sparsification that preserve positive definiteness, and a line-search with Sinkhorn fallback, yielding global convergence of Algorithm 2.
3. The numerical studies contain numerous simulations demonstrating its better convergence over other existing methodologies.

**Weaknesses:**

* Theorem 3 (Global Convergence) doesn’t specify its global convergence rate.
* There’s no theorem for the full PSN pipeline (Algorithm 3) that covers the switching logic (IP-EOT → Sinkhorn vs Newton). Theorem 3 is for Algorithm 2 only.
* Experiments could better evidence applicability through various examples (e.g., synthetic cases with known ground truth like Gaussians/uniforms).
* Compared to unregularized optimal transport problem [1], entropic optimal transport suffers from high space complexity as well as high time complexity and its representation is blurrier due to regularization term. This framework is also expected to suffer this issue.

Other issues
* The choice of numerical setting of the main hyperparameters (eg. total regularization $\beta$) is not discussed.
* It would be better if the experiment includes evidences showing its excellence in time complexity over other EOT methods( eg. time per iteration).

[1] Optimal Transport Barycenter via Nonconvex-Concave Minimax Optimization. Kim et al, 2025

**Questions:**

* In section 3, I would be glad if you clarify how using $\Delta\beta = \beta/l$ leads to numerical stability.
* Does this have a potential to be used for solving Wasserstein barycenter problem, as was in IPOT [2]?
* In section 4, I would appreciate if you literally explain how adjustments affect the global convergence. And can you specify the convergence rate for theorem3?
* Is this method scalable to high dimensional settings such as multivariate distributions?

[2]  A Fast Proximal Point Method for Computing Exact Wasserstein Distance. Xie et al. 2020.

---

> ### Author Response · Authors · 2025-11-22
> **Response to Reviewer nCyg (Weaknesses Part)**
>
> We thank the reviewer for their positive assessment of our work. We are particularly grateful for their recognition of our main contributions: the Inexact to Exact proximal scheme (Theorem 1) that ensures stability, the global convergence from Fast Sparse Newton method with Sinkhorn fallback, and the comprehensive numerical studies demonstrating superior performance.
>
> We thank the reviewer for the helpful suggestions, which have been incorporated into the revised version. We have revised the manuscript in accordance with all of your suggestions, and we have included a comparison between the previous and revised versions in the supplementary material.
>
> **W1\&Q3(b). (Global convergence rate)**
> Thank you for raising this question. Unfortunately, the classical Newton method does not, in general, possess a global convergence rate. In fact, standard Newton’s method is only guaranteed to achieve local quadratic convergence, and it lacks any form of global convergence guarantee.
>
> In our work, global convergence is ensured not by the Newton step itself, but by our specially designed line-search–based step-size strategy, which allows the algorithm to make progress even when the pure Newton step fails. This mechanism (detailed in Section 4.3 of the revised manuscript) enables us to establish global convergence, while still retaining the fast local convergence behavior of Newton’s method.
>
> **W2. (Full PSN pipeline theorem)**  Thanks for pointing out this omission. We have added theoretical guarantees for the full PSN pipeline (Algorithm 3) in the Section 5 of the revised version.
>
> **W3. (Various examples)**  Thanks for the helpful suggestion. We have evaluated the performance of PSN algorithm against existing methods on the ImageNet dataset in Appendix L of the revised version.
>
> **W4. (Compared to unregularized optimal transport problem)** We thank the reviewer for pointing out the recent progress on unregularized optimal transport (OT). WDHA [1] achieves a low per-iteration complexity of $O(n \log n)$ by leveraging the structure of the unregularized Wasserstein barycenter problem when distributions are discretized on a regular Euclidean grid. **This computational advantage of WDHA [1] is problem-specific and does not extend to general optimal transport settings**, where support points may be unstructured or non-uniformly distributed. In such cases, the entropically regularized OT formulation remains substantially more efficient—a key reason for the widespread adoption of Sinkhorn-based distances.
>
> We acknowledge that the regularized OT formulation tends to produce blurrier transport plans than the unregularized version. However, this is not necessarily a drawback. In many practical applications such as graph matching and feature selection, a certain level of smoothness introduced by the entropic regularization can actually improve robustness and generalization.
>
> **W5. (Hyperparameters)** We thank the reviewer for the valuable comment. The numerical settings of key hyperparameters—such as the total regularization $\beta$ and the marginal error threshold $\rho\_{\mathrm{th}}$—vary across different application scenarios, and thus no single configuration is universally optimal. To address this, we conducted sensitivity analyses by varying $\beta$ and $\rho\_{\mathrm{th}}$. Moreover, we have added discussions on $\Delta \beta$ and $\lambda$ in Appendix of the revised manuscript for your review. We hope that these clarifications address your concerns.
>
>
> **W6. (Time per iteration)** Thank you for the suggestion. We have included the time per iteration in the revised Appendix P to more clearly demonstrate the superiority of our method.

---

> ### Author Response · Authors · 2025-11-22
> **Response to Reviewer nCyg (Questions Part)**
>
> **Q1. (Stability from $\Delta \beta = \beta/\ell$)** Thanks for this nice question. When the full regularization $\beta$ is applied in one step and $\beta$ is large, the kernel $K=\exp(-C\beta)$ becomes numerically degenerate: most entries of $K$ to underflow to $0$.  This degeneracy directly destabilizes Sinkhorn scaling because the updates $u = a/(K v)$ and $v = b/(K^\top u)$ then require dividing by extremely small (or machine-zero) numbers, amplifying numerical noise and causing the scaling factors to blow up or collapse. Our IP-EOT use $\Delta\beta=\beta/l$ introduces the entropic regularization gradually, so each intermediate kernel remains well-conditioned, avoiding these overflow/underflow issues and ensuring stable Sinkhorn updates. We have added this clarification to Section 3 for a clearer presentation.
>
> **Q2. (Wasserstein barycenter)** Yes, it can be used for solving Wasserstein barycenter problem. Solving the Wasserstein barycenter problem fundamentally relies on repeatedly
> solving OT subproblems. Our method targets entropic OT, whose solution is a
> controlled approximation of the true OT solution, and the approximation level can be
> adjusted via the regularization parameter $\beta$. Therefore, because our algorithm provides a entropic OT solver, any task that IPOT can handle---including the Wasserstein barycenter problem---can also be accomplished by our method.
>
>
> **Q3(a). (Adjustments and the global convergence)**  Thanks for this nice question. The introduced adjustment—employing Sinkhorn iterations to escape plateau regions—enhances the algorithm's global convergence properties. Specifically, the step size is typically determined through a backtracking line search procedure to ensure objective function decrease at each Newton iteration. This approach starts with an initial step size and progressively shrinks it by a constant factor $ \in (0,1)$ (e.g., $ 0.5$) until sufficient objective improvement is achieved. However, a practical limitation arises when the line search yields excessively small step sizes, causing the algorithm to enter a plateau phase where objective values stagnate despite continued iterations, thus impeding convergence.
>
> In that stage, we can use Sinkhorn iterations to generate a robust direction ${p}\_k$, enabling escape from the stagnation region:
> $$f({z}\_k + {p}\_k) < f({z}\_k).$$
> Once a suitable step size $\alpha$ can be readily determined again, the method resumes Newton updates, thereby achieving both robustness and rapid convergence.
>
> Thank you for your question again, which motivated us to further improve this part of the manuscript. We have now separated this part into **4.3 Line Search with Sinkhorn Fallback** of the revised manuscript.
>
> **Q4. (Multivariate distributions)**
> Yes. The method naturally extends to high-dimensional (multivariate) settings because entropic optimal transport is dimension-agnostic. The mentioned imageNet tasks include 30-dimensional features. In entropic OT, the optimization is defined over discrete probability weights and a cost matrix $C\_{ij} = c(x\_i, y\_j)$, while the samples $x\_i, y\_j$ may lie in ${R}^d$ for any $d$.  The entropic regularization and Sinkhorn iterations do not depend on the data dimension, and only the cost computation changes. Therefore, multivariate distributions (e.g., images, point clouds, or high-dimensional embeddings) are fully supported without modification to the algorithm.
>
> In very high dimensions, the OT cost matrix $C$ may suffer from degraded discriminability due to the concentration of pairwise distances. This necessitates the use of low-dimensional projections or embeddings, which enable effective OT computation by working in a more amenable, lower-dimensional space.

---

### Note · Authors · 2026-01-27

I have read and agree with the venue's withdrawal policy on behalf of myself and my co-authors.

---

### Meta-Review · Area_Chair_nfmn · 2026-01-03

**Summary:**

The paper addresses the numerical solution of entropic optimal transport by proposing a hybrid solver designed to remain stable at large regularization and achieve high accuracy, with the main contribution being a two-stage Proximal-Sinkhorn-Newton (PSN) pipeline combining an IP-EOT warm start with Sinkhorn or sparse Newton refinement, together with convergence analysis and several practical stabilizers. Despite the relevance of the problem and the technical effort invested, several major weaknesses remain. First, while the authors substantially strengthened the theoretical side in the rebuttal by adding a convergence analysis for the full pipeline and clarifying the role of line search and Sinkhorn fallback, the absence of a global convergence rate and unquantified constants still limits the predictability and practical interpretability of the guarantees. Second, the empirical evaluation, although clearly expanded in the rebuttal with additional benchmarks (including ImageNet), larger-scale experiments, and improved baseline comparisons, remains only partially convincing, in particular, coverage across OT use cases is still limited. Third, the method appears to require careful and problem-dependent hyperparameter tuning; the rebuttal provides helpful sensitivity analyses and clearer guidance, but the overall tuning complexity remains a significant adoption cost. Finally, although many issues of rigor, definitions, and presentation were reportedly corrected in the rebuttal, the cumulative changes highlight that the paper in its current form required substantial clarification and restructuring. Overall, the authors made a serious rebuttal effort and addressed many reviewer concerns to a meaningful extent, but the need for an in-depth rewrite and a more definitive demonstration of scalability and positioning relative to prior work means that the paper cannot be accepted.

**Reviewer Concerns:**

Several substantive concerns raised by the reviewers were clearly addressed in the rebuttal. In particular, the authors responded carefully to questions about numerical stability, the role of the incremental regularization schedule, and the lack of guarantees for the full PSN pipeline by adding new theoretical results and clarifications. Many presentation issues, missing definitions, and ambiguities identified by multiple reviewers were also reportedly corrected, and the experimental section was expanded with additional baselines, sensitivity analyses, variance reporting, and larger-scale tests. However, a number of concerns remain outstanding. Theoretical guarantees still stop short of providing a usable global convergence rate or interpretable constants, which limits their practical value. Empirically, while scale and scope improved, the evaluation still does not fully convince with respect to general OT applications or competitive positioning against the strongest existing Newton-type solvers. Finally, although hyperparameter behavior is better documented, the overall complexity of tuning remains high and was not fully mitigated by the rebuttal.

**Reviewer Scores:**

- nCyg (initial score: 4): Likely no change. While their requests for a full-pipeline guarantee and clearer explanations were addressed, the absence of a convergence rate and limited application breadth would probably keep their assessment unchanged.
- C7nk (initial score: 4): Likely no change. Additional baselines, hyperparameter discussion, and clearer figures address many points, but the broader concerns about practical comparison and clarity likely remain.
- pAcV (initial score: 4): Likely no change. Scalability experiments and hyperparameter analyses directly target their concerns, yet the remaining questions about robustness and real-world breadth suggest their score would stay stable.
- QcbC (initial score: 6): Possible slight decrease or no change. While hyperparameter intuition and convergence clarifications were added, the lack of stronger empirical or theoretical closure may temper their relatively positive stance.
- YPFa (initial score: 4): Likely no change. Most issues on rigor, convergence of the hybrid method, and experimental variance were addressed, but the overall contribution level probably remains similar in their view.
- MkrW (initial score: 2): No change. Despite added experiments and clarifications, this reviewer explicitly remained unconvinced about scalability and comparative strength.

---

### Decision · Program_Chairs · 2026-01-26

Reject